
# Observing Entrainment Mixing, Photochemical Ozone Production, and Regional Methane Emissions by Aircraft Using a Simple Mixed-Layer Model

Justin F. Trousdell[1], Stephen A. Conley[1,2], Andy Post[*], Ian C. Faloona[1]

[1]Department of Land, Air, and Water Resources, University of California Davis, United States
[2]Scientific Aviation, Inc., Boulder, Colorado, United States
[*]now at the California Air Resources Board, Sacramento, California, United States

*Correspondence to*: Ian C. Faloona (icfaloona@ucdavis.edu)

**Abstract.** In situ flight data from two distinct campaigns during winter and summer seasons in the San Joaquin Valley (SJV) of California are used to calculate boundary layer entrainment rates, ozone photochemical production rates, and regional methane emissions. Flights near Fresno, California in January and February 2013 were conducted in concert with the NASA DISCOVER–AQ project. The second campaign (ArvinO3), consisting of eleven days of flights spanning June through September 2013 and in June 2014 focused on the southern end of the SJV between Bakersfield and the small town of Arvin, California, a region notorious for frequent violations of ozone air quality standards. Entrainment velocities, the parameterized rates at which free tropospheric air is incorporated into the atmospheric boundary layer (ABL), are estimated from a detailed budget of the inversion base height. During the winter campaign near Fresno, we find an average midday entrainment velocity of 1.5 cm s$^{-1}$, and a maximum of 2.4 cm s$^{-1}$. The entrainment velocities derived during the summer months near Bakersfield averaged 3 cm s$^{-1}$ (ranging from 0.9 – 6.5 cm s$^{-1}$), consistent with stronger surface heating in the summer months. Using published data on boundary layer heights we find that entrainment rates across the Central Valley of California have a bimodal annual distribution peaking in spring and fall when the lower tropospheric stability (LTS) is changing most rapidly. Applying the entrainment velocities to a simple mixed–layer model of three other scalars ($O_3$, $CH_4$, and $H_2O$), we solve for ozone photochemical production rates and find wintertime ozone production ($2.8 \pm 0.7$ ppb h$^{-1}$) to be about one-third as large as in the summer months ($8.2 \pm 3.1$ ppb h$^{-1}$). Moreover, the summertime ozone production rates observed above Bakersfield/Arvin exhibit an *inverse* relationship to a proxy for the VOC:NO$_x$ ratio (aircraft [$CH_4$] divided by surface [$NO_2$]), consistent with a NO$_x$–limited photochemical environment. A similar budget closure approach is used to derive the regional emissions of methane, yielding 100 Gg yr$^{-1}$ for the winter near Fresno and 170 Gg yr$^{-1}$ in the summer around Bakersfield. These estimates are 3.6 and 2.4 times larger, respectively, than current state inventories suggest. Finally, by performing a boundary layer budget for water vapour, surface evapotranspiration rates appear to be consistently ~55% of the reference values reported by the California Irrigation Management Information System (CIMIS) for nearby weather stations.

## 1 Introduction

During the daytime over the continents, when ozone ($O_3$) reaches its peak, convective thermals generated at the surface rise and penetrate into the stable layer that demarcates the interface between the turbulent boundary layer and the laminar (non–turbulent) free troposphere (FT) above it. The continuous action of these thermals penetrating into the laminar overlying air and falling back into the boundary layer gives rise to an irreversible mixing process that causes the layer to grow up through the mid–



morning to afternoon, diluting the air in the atmospheric boundary layer (ABL) with that from the free troposphere. The overall process is referred to as entrainment, and when the two layers contain different amounts of any scalar quantity (e.g. ozone concentration, water vapour, enthalpy), this mixing process tends to be a significant contributor to the ABL budget of the scalar, and therefore vital to predicting and interpreting its abundance at the surface.

Typically entrainment is not treated explicitly in chemical transport models because the scales of motion, taking place predominantly within the ABL capping inversion, are suppressed in vertical extent due to the thermodynamic stability of this layer. Consequently the mixing tends to be sub–grid in nature and requires some form of parameterization. Many aircraft measurements of this parameter have been attempted using the tracer method (Nichols, 1984; Kawa & Pearson, 1989; Faloona et al., 2005; Karl et al., 2013), however this requires the use of eddy correlation to measure the turbulent fluxes near the base of the
ABL inversion.  Because the aircraft in the present study, operated by Scientific Aviation, Inc. does not currently have the capability to measure vertical wind speeds, we use here instead a budget of the inversion base height (Wood & Bretherton, 2004; Faloona et al., 2005; Albrecht et al., 2016) to infer the entrainment rate.

Another meteorological process that can strongly influence surface concentrations is mesoscale advection, and owing to the complexity of the surface wind field in complex terrain and heterogeneity of surface sources of trace gases, this term has
traditionally been difficult to account for in ground–based air pollution studies. By way of targeted airborne campaigns we are able to probe the regional ABL vertically and horizontally and calculate entrainment rates and mesoscale advection. This flight strategy thus helps to illuminate the origins of surface levels of various chemical constituents, by comparing the observations, including the rate of advection, with the overall scalar budgeting equation. Past measurements of DMS, $SO_2$, and $O_3$ budgets carried out over the presumably homogenous ocean indicate that while on average the advection term is not large, it can be
dominant on any given day, and so must be considered when looking at individual episodes (Conley et al., 2009; Faloona et al., 2010; Conley et al., 2011).

Outlined in the seminal work of Lenschow et al. (1981) are original applications of the scalar budgeting techniques used by Warner & Telford (1965) and Lenschow (1970) to help validate the newly developing technique of eddy covariance for measuring sensible heat fluxes by aircraft. Lenschow et al. (1981) go on to describe the effectiveness of well-designed aircraft
ABL studies in determining the net source or sink (in their case for ozone) given the careful measurement of the other dynamically controlled terms. The technique can be generalized to any scalar budget (i.e. ozone, water vapour, DMS, $SO_2$) to enable the calculation of important residuals including source or sink terms for non–conserved species (Kawa & Pearson, 1989; Bandy et al., 2012; Conley et al., 2009; Faloona et al., 2010). In the process of quantifying the individual terms of the budget equations, their relative importance can be weighted to provide a better understanding of the leading causes and factors affecting
surface concentrations.

A contemporary challenge for air quality monitoring in the age of increasing sophistication of remote sensing from space is correlating surface concentrations of key trace gases (ex. $NO_x$, $O_3$, etc.), with column measurements from satellite. Many air pollutants of interest are concentrated predominantly in the boundary layer, where the main sources are often located, thus there is a strong need for understanding the diurnal behaviour of the mixed layer. One possible way to improve the correlation between
surface and column concentrations is by understanding its connection to ABL height, and also the role of ABL mixing with the FT (entrainment). The depth of the ABL directly affects the concentration of tracers (i.e. surface levels), as they will be diluted and mixed throughout it. Recent studies in California by Al−Saadi et al. (2008) suggest that lidar measurements of ABL height





can normalize column observations of AOD (Aerosol Optical Depth) to greatly improve correlations to surface PM2.5 (Particulate Matter up to 2.5 micrometers in size). Improving the inference of surface concentrations from satellite data is among the chief scientific goals of the NASA experiment DISCOVER–AQ (Deriving Information on Surface Conditions from COlumn and VERtically Resolved Observations Relevant to Air Quality). Seven of our flights were conducted during the California

campaign of DISCOVER–AQ, in an effort to support their scientific mission. DISCOVER–AQ sought to use concurrent integrated observations to meet this goal, among them was the University of California Davis (UC Davis) in situ aircraft measurements of trace gas, and thermodynamic budgets to better understand the diurnal behaviour of the wintertime boundary layer in the San Joaquin Valley.

The San Joaquin Valley (SJV) of California is well known for its air quality challenges. As of 2013 the Valley is a non-

attainment site for the state standard and the federal 8-hour standard for $O_3$, a status that is only going to be aggravated by the recent reduction in the federal 8h standard to 70 ppbv (US EPA). Additionally, the majority of the SJV, especially the southern portion, is designated non-attainment for PM2.5 for state and federal standards (California Air Resources Board (CARB)) as of 2013. In winter the SJV is plagued by PM2.5 problems relating to temperature inversions, low mixed layer heights, and more recently extreme drought conditions. In the southern SJV weak surface winds and a unique basin topography add to the problem

of stagnation. Although more dominant during the summer, on average a strong temperature inversion exists aloft over the SJV restricting vertical motions and venting of pollution. High temperatures in summer coupled with an effective combination of $NO_x$ and VOC's leads to high photochemical production of $O_3$ in the SJV. From 2003 to 2005 the SJV saw on average 105 exceedances of federal 8h $O_3$ from 20 monitoring sights (San Joaquin Valley Air Quality District Ozone Air Quality Plan 2007).

Here we will discuss the results of the two flight campaigns targeting the SJV, and the efficacy of using calculated regional

entrainment rates with the addition of known budget terms in exposing important residuals like methane emissions and in situ $O_3$ photochemical production. First we will start by discussing the uniqueness of the flight regions including the synoptic setting as well as the important mesoscale features. The next section will detail our mixed layer model, followed by the results from the analysis including budget residuals and finally the conclusions of this work.

## 2   Experimental Description

### 25   2.1   Synoptic and Geophysical Settings

The arid weather experienced throughout most of California during the summer is under the weight of the prevailing Pacific High, centred near 35° N some 2000 km offshore (Fig. 1 bottom right), which blocks storm systems from hitting the state instead shunting them northward towards Canada. The domineering anticyclone moreover drives synoptic scale subsidence on its downwind flank over the region. A strong thermodynamic "lid" or temperature inversion is set up by the synoptic subsidence,

resisting convective motions throughout the lower atmosphere. This is why the state experiences a long dry, hot and sunny summer. The zonal pressure gradient and surface friction impel a degree of onshore flow (atmospheric Ekman transport) that is principally blocked by the coastal mountains. The low–level summertime airflow into the interior of the state is therefore restricted to the main break in the Coast Range around the San Francisco Bay area and is strengthened by the land–ocean thermal contrast, with air entering the Carquinez Strait just beyond the San Francisco Bay and diverging into the conjoined valleys

(Schultz et al. 1961; Frenzel 1962; Hays et al. 1984; Moore el al. 1987; Zaremba and Carroll 1999). The air is diverted northwest into the Sacramento Valley and southeast into the San Joaquin Valley as it butts up against the tall Sierra Nevada mountain range





on the far side of the Central Valley and large scale vertical motions are generally suppressed due to the stable subsidence inversion above.

In the SJV a nonlinear superposition of flows dictates the observed winds. In addition to the synoptic forcing discussed above, there is a direct thermal forcing of the mountain-valley circulation with consequent up-slope flows inducing mesoscale

subsidence over the central valley floor (Rampanelli et al., 2004; Shcmidli & Rotunno, 2010). In the far southern end of the San Joaquin up-valley air is forced to converge as it runs into the steep topography of the Tehachapi Mountains. This low-level orographic convergence, which was shown in ABL wind data by Bianco et al. (2011), gives rise to mesoscale uplift especially pronounced at the cul-de-sac of the valley. Monthly composites of vertical velocity (omega) from the National Center of Environmental Prediction/North American Regional Reanalysis (NCEP/NARR) dataset, averaged over the decade from 2004 to

2013, are depicted in Figure 1. Upward motion is present across large swaths of the Central Valley during summer, likely due to orographic lift on the windward side of the Sierras, but it appears especially strong in the southern end of the valley (Fig. 1) where the thermal valley wind and southern mountains augment the effect.

As the evening progresses down–slope, katabatic flows, which are strongest on the flanks of the highest parts of the Sierra Nevada range to the east, interact with the valley wind, taking the form in the evening of a low–level jet, and giving rise to a

mesoscale, cyclonic circulation referred to as the Fresno Eddy (Lin & Jao, 1995; Zhong et al., 2004; and Bao et al., 2008). The Fresno Eddy is most pronounced in the early morning as a combinatory effect of the up–valley low–level jet, which peaks before midnight and the southeasterly down-slope flow strengthening throughout the night until sunrise.

Seven flights from 16 January to 4 February of 2013 were deployed across the San Joaquin valley transverse to its axis with extensive vertical profiling of the ABL and the free troposphere (FT) above it, in conjunction with the NASA DISCOVER–AQ

California campaign (flight region 1, see Fig. 2). In each vertical profile up and down through the ABL we monitored the inversion height in addition to a suite of scalar measurements (ozone, water vapour, methane, horizontal winds, carbon dioxide, and temperature). In addition, on each profile we fly up through the ABL top in order to characterize the composition and thermodynamic properties of the FT: measuring the difference in a scalar quantity from a well–mixed layer to the FT is essential to understanding entrainment fluxes. The second set of deployments was focused at the southern end of the SJV during the

summer months employing a slightly different flight strategy (flight region 2, see Fig. 2). Although vertical probing up and out of the ABL was consistent, a greater emphasis was placed on the horizontal extent of the measurements in the direction of the mean ABL wind. The main focus of this campaign was to better understand the cause of the large number of ozone NAAQS exceedances in this region surrounding the small town of Arvin. To do so required a thorough quantification of the horizontal advection as well the entrainment flux of $O_3$ (directly related to entrainment rates). Flights were targeted at $O_3$ exceedance

episodes with each of four deployments lasting 2–3 days spanning two summers (2013-2014) between June and September.

The complex mesoscale terrain plays an important role in the valley atmosphere, and needed to be taken into consideration for this study. The influence of topography on the thermally driven flow pattern arising from land–ocean contrast in the California Central Valley during the summer is discussed in Zhong et al. (2004). Their study employed the use of 22 wind profilers with radio acoustic sounding systems (RASS) to vertically probe the atmosphere. The authors suggest, based on temperature profiles

in the lowest 800 m, that the mixed layer height, which probably exceeds 1000 m AGL, slopes up valley in the San Joaquin. Additionally, the thermally driven flow pattern frequently extends upward to 800–1000 m AGL. Bianco et al. (2011), investigating various factors influencing ABL height in the Central Valley, reported low–level convergence in the southern end





of the valley leading to increased ABL heights. They did so by looking at the difference in up–valley wind between two sites in the SJV, Chowchilla and Lost Hills. This is in contrast to sites to the north in the SJV which see a shoaling in the summer months, likely due to cold air advection from the coast, or subsidence induced from the valley flow (far from the cul-de-sac at the Tehachapi Mountains), or possibly other causes such as land use, wherein different irrigation patterns may lead to a different

partitioning of latent and sensible heat fluxes. Our study corroborates the convergence in the southern end of the valley in that the NCEP/NARR reanalysis data set shows strong uplift at the southern extremity of the SJV, and that there is often an unmistakable decrease in meridional winds approaching the southern mountains observed by the aircraft winds (data not shown.)

The SJV is flanked by three mountain ranges: the Sierra Nevada mountain range to the east, the Tehachapi mountain range south, and the Pacific Coast Ranges to the west. Winds within a mountain–valley system are observed to exhibit a diurnal pattern; up–

slope and up–valley flow during the day and down–slope and down–valley flow at night. Drivers for mountain–valley flows are usually understood to be thermal gradients between the atmosphere near the valley floor, above the valley floor and the atmosphere adjacent to the mountain sidewalls. Valleys with varying width or a sloped valley floor develop along-valley winds, which can be sufficiently described by a hydrostatic pressure difference between a narrow section of the valley compared to a wider one, described as a plain. This difference in hydrostatic pressure is a result of the valley volume effect: the valley air

volume is smaller yet it still receives the same amount of insolation as the plain, with all other factors taken to be equal, leading to higher average temperatures in the valley (Whiteman 1990). The valley volume argument holds only when there is no transport up and out of the valley, which has been shown not to hold in general, but nonetheless this technique is a good qualitative explanation of the thermally induced valley wind (Schmidli and Rotunno 2010). In the SJV, the strong up–valley wind is generally linked to the large thermal gradient between the interior valley and the air at the coast, so the valley wind is

augmented by the sea breeze. The coastal air enters through the Carquinez Strait and bifurcates north and south from the delta region. Large scale down–valley (southeasterly) flow is generally not observed in the SJV but a significant reduction in the strong up-valley (northwesterly) flow is observed at night The up-valley flow peaks just before midnight and reaches a minimum in the mid-morning hours. The maximum is reached when the ageostrophic wind component aligns with up-valley flow before midnight, the result of an inertial oscillation (Zhong et al. 2004) in response to the decoupling of ABL winds from the surface

after sunset. Zhong et al. (2004) estimates the phase difference, or response time, between the along valley wind and the along valley pressure gradient in the SJV to be about four hours. This helps explain why the up–valley flow is strongest at night yet the peak in pressure difference is in the late afternoon.

## 2.2 Aircraft Measurements

Our flight data was collected aboard a single engine Mooney TLS, operated by Scientific Aviation, Inc.

(http://www.scientificaviation.com), and piloted by one of the authors (SC). The Mooney is outfitted with a 2B Technologies $O_3$ monitor, a Vaisala HMP60 temperature and Relative Humidity probe, a modified Picarro 2301f Cavity Ring–Down Spectrometer (CRDS) to measure $CO_2$, $CH_4$, and $H_2O$, and an Aspen Avionics PFD1000 flight display delivering pressure, altitude, true air speed, etc. Measurement of the horizontal wind is accomplished using a novel technique developed for easy and inexpensive deployment on a single engine aircraft. Utilizing a dual GPS antenna to provide accurate airplane heading and a

ground velocity by vector subtraction from true air speed (TAS) the horizontal wind is calculated, a technique outlined in Conley et al. (2014).



### 2.3 Sortie Strategies

In order to support the objectives of the DISCOVER–AQ campaign by probing the boundary layer dynamics near the northern edge of the domain, the aircraft was flown back and forth perpendicular to the valley axis approximately between the NASA profile stations at Fresno and Tranquility (Fig.2). In the absence of making fast vertical wind measurements, we derive entrainment rates in a novel way using a complete scalar budget of the ABL height throughout each flight targeted from midday to late afternoon hours (usually 11:00–16:00 PST). The flight hours are specifically chosen to focus on the ABL dynamics after its initial, rapid growth through the residual layer in the mid-morning. The inferred entrainment rates derived from the ABL height–budget, are then used in all of the scalar budgets to reveal $O_3$ photochemical production rates, surface latent heat fluxes, and regional methane emissions as residuals.

To study the processes that govern the evolution of the surface concentration of $O_3$ during the summer months in the southern SJV more in-depth, we performed an airborne experiment in collaboration with Scientific Aviation, Inc. targeting the vicinity of Arvin, California during the summers of 2013 and 2014. Flying around and upwind of Arvin 3–7 hours per day during each of the four 3 day campaigns, observations of wind, temperature, methane, water vapour, and ozone were used to measure the principal dynamical components of the total ozone budget: namely, advective up–valley transport within the ABL and entrainment mixing from above. By comparing these measured dynamical terms with the observed $O_3$ rise throughout the region during the afternoon, and using a reasonable parameterization of dry deposition, the net photochemical production rate can be inferred. Consequently, the relative contributions of these processes to the resulting surface $O_3$ concentration can be estimated for midday conditions, which are most important in determining whether an ozone exceedance of the NAAQS is reached. On one of the flights during the second deployment (15 August 2013) we additionally made $NO_2$ measurements with a Los Gatos Research cavity enhanced absorption spectrometer. All flights, for both campaigns, targeted days with weak horizontal winds in the ABL because stagnation tends to accompany both wintertime PM2.5 and summertime $O_3$ episodes.

### 2.4 NARR Data

Because we are not able to accurately measure mean vertical wind speeds by aircraft currently, we resort to the NCEP NARR dataset to estimate the mean vertical wind speed at the top of the ABL during each flight. NARR is an extension of the NCEP global reanalysis, and was created to provide long–term consistent climate data focused over the U.S. at a regional scale. The model runs at 32 km resolution with 45 vertical layers providing data eight times a day with a reanalysis period from 1979–2015. More information about this reanalysis data set can be found at http://www.esrl.noaa.gov/psd/data/gridded/data.narr.html.

### 2.5 NOAA Sounding System Data

We make heavy use of the data collected by NOAA during 2008 from five 915 MHz radar wind profilers equipped with radio acoustic sounding systems (RASS) distributed across the Central Valley and reported in Bianco et al. (2011). Briefly the radio signal backscatter is augmented in regions with strong fluctuations in temperature and water vapour as exists in the entrainment zone at the top of the ABL. The method of Bianco et al. (2008), which uses not only the backscattered intensity, but further includes the vertical velocity variance and its spectral width to automatically select the ABL top throughout the day. The minimum gate height for these profilers is 120–140 m above ground, and their vertical resolution is 60 m. To evaluate the average ABL growth rates we simply subtract the mean height at 11:00 from 15:00 and divide by the 5 hour interval, as per the Fundamental Theorem of Calculus.





### 2.6    Budget of the ABL Inversion Height

Quite often the growth rate of the boundary layer is interpreted as equivalent to the entrainment velocity or volume flux of FT air into the ABL (Tennekes 1973), assuming that there is no large scale mean vertical wind. However, in most situations the ABL growth ($\frac{dz_i}{dt}$) is actually determined by the difference of two distinct processes: the entrainment which is considered to be driven by micrometeorological factors (viz. surface buoyancy flux, inversion strength, and possibly wind shear across the inversion), and the larger scale subsidence in the lower FT just above the ABL ($W_{z_i}$)

$$w_e = \frac{dz_i}{dt} - W_{z_i} \tag{1}$$

In a seminal paper on the effects of surface heating on the inversion height, Ball (1960) declared that there are several processes that counteract the tendency of entrainment to raise the inversion height. One is that "horizontal divergence in the lower layers, accompanied by subsidence at inversion level, may be sufficient to counteract the rise", and the other is that the "inversion usually slopes upward along the trajectories and thus advection tends to lower the inversion at a fixed point." To be even more precise we consider the total derivative of the ABL or mixed layer height ($z_i$) and expand it into the Eulerian derivative of ABL height and an advection term. The resultant $z_i$ budget equation leads to a relationship between the entrainment velocity, the observed local ABL growth rate, the mean advection of ABL depth, and the mean vertical velocity at the inversion height:

$$w_e = \frac{\partial z_i}{\partial t} + U \frac{\partial z_i}{\partial x} - W_{z_i} \tag{2}$$

The first two terms on the right hand side of Eq. (2) are, in principle, easily observed by aircraft, while the last term has evaded careful measurement by aircraft or any other means (Lenschow et al., 1999; Angevine 1997; Lenschow et al., 2007). While the sorties provided a sufficient number of ABL crossings to estimate the ABL growth rate with acceptable accuracy, there were generally not enough at different locations to capture an unbiased, two-dimensional gradient of the inversion height (second term on the rhs of Eq. 2). Consequently, we estimate the advection term using the gradient in ABL height as determined from the NCEP/NARR data set in conjunction with the observed in situ mean wind (Fig. 3). The observations of $z_i$ indicate that the reanalysis data does not predict the absolute boundary layer depth with great accuracy in the Central Valley. This is most likely due to the fact that the model does not treat the heavily irrigated agricultural land surface with any fidelity. Inspection of the surface latent heat fluxes in the model (data not shown) indicate unrealistically small values for a region with such fecund agricultural productivity. Nevertheless, we assume here that the reanalysis data does capture the gradients of ABL depth reasonably well. In fact, the gradients evinced in Fig. 3 are in rough accord with those reported in Bianco et al. (2011), approximately 500 m difference across the lower ~200 km of the southern SJV. The large–scale vertical mean wind, $W$, is derived from the NCEP/NARR pressure velocity omega ($\omega = \frac{dp}{dt}$), and the surface pressure tendency neglecting horizontal pressure advection and assuming hydrostatic balance:

$$W = \frac{1}{\rho g} * \left( \omega - \frac{\partial p}{\partial t} \right), \tag{3}$$

The pressure level from which to select the omega value was chosen using the hypsometric equation $\left( p_2 = p_1 * \exp\left(-\frac{z_i * g}{R_d * \overline{T}}\right) \right)$ using an average observed ABL height, $z_i$, an average ABL temperature, $\overline{T}$, for the flight duration, $R_d$ is the dry air gas constant, and an estimated average surface pressure, $p_1$, of 1010.5 mb for June–Sept, and 1020 mb for Jan–Feb.



The local pressure change is estimated by the surface pressure tendency using hourly data from several CARB (The California Air Resources Board: http://www.arb.ca.gov/aqmis2/metselect.php) meteorology stations in the area over the flight time. Throughout the afternoon during both seasons the valley experiences a fairly strong and consistent drop in surface pressure of approximately 0.6 mb h$^{-1}$. Similar diurnal oscillations of surface pressure were found by Li et al. (2009) to be prevalent in deep mountain valleys of the western US. Although these pressure changes are large by synoptic standards, they are generally an order of magnitude smaller than the omega values.

### 2.7 Mixed Layer Model

Briefly, the ozone budget of the ABL can be mathematically represented as:

$$\frac{\partial O_3}{\partial t} = -U\frac{\partial O_3}{\partial x} - \frac{1}{z_i}\left(v_{dep}O_3 + F_{ent}\right) + P \,, \tag{4}$$

The first term on the left represents the observed temporal trend in a fixed region, the second term represents the advection (the influence of the mean wind, $U$, acting on the large scale gradient in the $O_3$ field), $z_i$ is the ABL depth, $v_{dep}$ is the deposition velocity representing foliar uptake of $O_3$ by plants at the surface, $F_{ent}$ is the entrainment flux due to mixing at the top of the ABL, and $P$ represents the net photochemical production (Conley et al., 2011).

In these types of ABL budget studies, a common parameterization for the entrainment flux is employed that estimates it to be proportional to the difference in concentration between the ABL and the free troposphere, across the inversion interface (see Fig. 7 for an example). In the ArvinO3 study, the budget for water vapour, a conserved quantity in the absence of precipitation, was initially used to infer the entrainment flux (similar to the case of the $O_3$ budget, entrainment is a dilution or drying effect.) Based on the observed difference in water vapour between the ABL and the FT, the entrainment velocity is derived. In principle this entrainment velocity is applicable to any scalar quantity (e.g., potential temperature, water, ozone, or methane) and represents a net consequence of the turbulent action in the ABL. In strongly convective situations such as commonly experienced in the summertime Central Valley during the day, the vigour of entrainment is a balance between the surface heating, which generates thermals that overshoot the ABL top, opposed by the strength of the temperature inversion resisting their vertical penetration. The relationship linking the entrainment flux, $F_{ent}$, entrainment velocity, $w_e$, and the jump in concentration between the ABL and FT of some scalar is:

$$F_{ent} = w_e * \Delta[C]_{(FT-ABL)} \tag{5}$$

The scalar jump ($\Delta[C]_{(FT-ABL)}$) is usually negative for a compound with a surface source (e.g. water, methane, and ozone), and a positive entrainment velocity holds for a turbulent boundary layer. Therefore, the entrainment flux is negative, meaning entrainment of FT air is diluting the ABL air's concentration. Initially the water vapour budget equation was chosen to calculate the entrainment velocity. However, difficulties were met when trying to make good estimates for regional surface latent heat fluxes in order to close the budget equation. In the absence of clouds and precipitation the water vapour budget equation is:

$$\frac{\partial q}{\partial t} = -U\frac{\partial q}{\partial x} + \frac{\overline{(w'q')}_s - \overline{(w'q')}_{z_i}}{z_i} \tag{6}$$

The third term here is simply the first order flux divergence for water vapour in the ABL. The next term $\overline{(w'q')}_{z_i}$ is the entrainment flux, which is normally positive, upward, meaning drying by the FT air dilution. The surface latent heat flux, $\overline{(w'q')}_s$, was taken from the NCEP/NARR dataset, which we find significantly underestimates the true value in the region of





interest. We then looked to the reference evapotranspiration ($ET_o$) at various sites throughout the Central Valley from the California Irrigation Management System (CIMIS). Evapotranspiration is directly related to latent heat flux by way of the latent heat of vaporization ($L_v$). $ET_o$ comes from standardized grass or alfalfa over which the measurement stations are situated, and it includes loss of water from the soil and plant surfaces. Although agriculture is prevalent in the area of interest it does not

represent the entire land surface. Using this value, $ET_o$, should however represent an upper limit for the latent heat flux of the region. Ultimately the approach using the budget of boundary layer inversion height, outlined in Section 2.6 was taken to calculate the entrainment rate.

## 3 Discussion of Results

Below we discuss the various important results that can be extracted from a flight strategy targeting a fixed region of 50–100 km

scale, and carefully tracking the changes in thermodynamic and chemical properties of the air mass. Because the sampling specifically targets the time of day when the boundary layer is actively entraining from the FT (excluding its initial phase of 'encroaching' through the residual layer), all of the results for entrainment rates, surface emissions of methane, evapotranspiration, and in situ photochemical production, all pertain to the 5 hour period of late–morning to early afternoon from 11:00 to 16:00 local standard time.

### 3.1 ABL Growth and Entrainment Rates

The airborne data measuring ABL growth rates are used to diagnose the entrainment rate by budgeting of $z_i$ as expressed in Eq. (2). ABL heights and their diurnal changes are shown for all the flights in Fig. 4 compared with the corresponding RASS data presented in Bianco et al. (2011). The Chowchilla site is 50 km upwind from Fresno, and the Lost Hills site is just on the upwind edge of our sampling domain for the ArvinO3 study (Fig. 2). Both the boundary layer depths and their growth rates measured in

the airborne experiments appear to be slightly lower than the Bianco et al. (2011) seasonal averages. The discrepancy is probably attributable to both airborne experiments specifically targeting the stagnation, high–pressure synoptic settings that characterize both the wintertime PM2.5 and summertime ozone episodes, which in principle suppress ABL development due to subsidence. Table 1 summarizes the estimated entrainment velocities from the two experiments, indicating a range between near zero (or below our detection limit of about 0.5 cm s$^{-1}$) to 2.4 cm s$^{-1}$ during the wintertime in the central SJV (average of 1.5 ± 0.9 cm s$^{-1}$),

and approximately 0.9 cm s$^{-1}$ to 6.5 cm s$^{-1}$ during the summertime over the southern SJV (average of 3.0 ± 2.1 cm s$^{-1}$.) Broadly comparable values have been observed in other continental studies: 4.3 cm s$^{-1}$ during late July over grassland in the Netherlands in a study by de Arellano et al. (2004), 1.4 ± 0.3, 5.5 ± 1, and 9.6 ± 1.5 cm s$^{-1}$ over the foothills of the Sierra Nevada adjacent to the California's Central Valley using isoprene flux measurements during June by Karl et al. (2013), and 5 ± 1 cm s$^{-1}$ over the Ozark mountains in the southeastern U.S. during September by Wolfe et al. (2015). As far as we can tell, the data presented here

are the first of their kind to estimate entrainment during the winter season, which although observed to be smaller as expected because of weaker surface heating, are critical to understanding the meteorological influence on the valley's PM2.5 episodes.

The entrainment velocities estimated in the two studies show evidence that they are linked to physically relevant surface parameters present during the flights. For example, the summertime entrainment velocities correlate well with the average ABL potential temperature ($r^2$ = 0.57, data not shown), insinuating that the forcing that heats the boundary layer (surface and

consequent entrainment heat fluxes) is intimately linked to the entrainment rate. In a similar vein, the wintertime entrainment velocities correlate well with estimates of net surface radiation found in the NARR data set ($r^2$ = 0.68, data not shown.) A



climatology of boundary layer heights reported by Pal & Haeffelin (2015) near Paris showed that although surface heat fluxes should most directly control the boundary layer height, a better correlation was found, on diurnal to seasonal time scales, with the surface down-welling shortwave radiation. While surface fluxes were not directly measured as part of the experimental set up, we turn to the surface solar radiation measured with pyranometers by the CIMIS network across the region. Figure 5 shows a

very strong linear relationship with the surface pyranometer observations and the average boundary layer height for each flight. In fact, the linear fits for each separate experiment seem to be the same within the uncertainties of the fits, and the slopes of 1.5 m $(Wm^{-2})^{-1}$ are similar to those reported in Pal & Haeffelin (2015) of 1.7 m $(Wm^{-2})^{-1}$

Pal & Haeffelin (2015) further survey nearly a dozen past studies that reported ABL growth rates over different seasons ranging from 0.8 to 8.3 cm s$^{-1}$. There are two main reasons that these growth rates are not exactly comparable to the entrainment

velocities reported here. First, most of these studies do not explicitly take into account horizontal or vertical $z_i$ advection (the last two terms in Eq. (2)). Second, the convention used by many is to report ABL growth rates for the interval from when the surface heat flux reverses sign shortly after sunrise to the time when the boundary layer height is 90 % of its daily maximum. Such growth rates are thus a combination of the rapid growth through the nearly statically neutral residual layer in the morning and the slower growth near midday when the ABL is actively entraining air from the free troposphere. For the chemical budgets under

consideration in this work, we contend that it is more important to quantify the late morning to early afternoon entrainment mixing between the ABL and FT. Entrainment of the residual layer in the early morning (sometimes called fumigation) represents merely a recycling of the previous day's boundary layer air (albeit from sources at a remove of an overnight advection scale of order 100 km). For the purposes of estimating regional source strengths or regional in situ photochemistry, we suggest that the more pertinent mixing process is the dilution of the anthropogenically influenced ABL air mass by the more global

'baseline' FT air. Both of these differences lead to the realization that the ABL growth rates reported by Pal & Haeffelin (2015) and references therein, should be systematically larger than the entrainment velocities reported in this study, at least under fair weather conditions (subsidence). Our data from the DISCOVER–AQ wintertime study presented in Table 1 indicates that the advection and subsidence terms may not be first order, especially for longer period averages, and therefore may be comparable to other ABL growth rate statistics reported in the literature. This conjecture is consistent with conclusions from previous budget

studies indicating that while advection may make a significant contribution to the scalar budget on any specific day, it may average out when considered on longer intervals (Conley et al., 2009; Faloona et al., 2010). A similar argument can be made for the vertical velocity term in Eq. (2); namely, that it may average to near zero across periods of instability and uplift, and periods of fair weather and subsidence. In a similar vein, the average $z_i$ budgets for the southern SJV (ArvinO3 in Table 1) show a sizeable average orographic uplift and opposing horizontal advection of $z_i$, which together may be in a quasi-steady state nearly

cancelling over long periods of weeks to months.

It follows that although not exactly equivalent to entrainment as described by Eq. (2), the range of ABL growth rates reported in the literature (from Pal & Haeffelin, 2015, and references therein) is nonetheless reasonably consistent with the data reported in Table 1. In the studies that report both winter and summer seasonal average ABL growth rates (Chen et al., 2001; van der Kamp & McKendry, 2010; Lewis et al., 2013; Schween et al., 2013; Korhonen et al., 2014; and Pal & Haeffelin, 2015) the summer to

winter ratios tend to range from 1.4–3.0, with an average of 2.0. This is consistent with our results indicating entrainment rates 80 % higher in the SJV during summer than winter.

Bianco et al. (2011) postulate that convergence at the southern end of the SJV in summer leads to deeper ABLs there than in other parts of the valley, closer to the delta inflow region, which are influenced by strong marine layer inflow. A typical slope of





ABL height up the SJV from Bianco et al. (2011) can be estimated from the Chowchilla and Lost Hills sites, which differ by about 750 m (from their Fig. 2) over a distance of approximately 175 km for the summer months. Applying to this gradient a calculated average along–valley wind at the top of the ABL of 2.5 m s$^{-1}$ gives an advection term of -1.1 cm s$^{-1}$. This estimate compares well to the -1.15 cm s$^{-1}$ reported in Table 1, derived from the NARR data set from our flight region 2 during summer.

In addition, Bianco et al. (2011) make a rough estimate of convergence in the southern SJV by simply taking the difference in the horizontal along–valley wind at the two sites (2.5 m s$^{-1}$ between Jun-Sep), divided by the distance between them, leading to ABL flow convergence of 1.4x10$^{-5}$ s$^{-1}$. Such convergence would lead to an uplift of 1.4 cm s$^{-1}$ at the top of a typical 1000 m boundary layer. This estimate is, again, right inline with our estimates from this study. From our findings it appears that the local time rate of change of observed ABL height nearly matches the entrainment rate when both are averaged over all the summer flights. The convergent uplift and the advection of ABL height appear to balance on average in the southern SJV. This suggests that the radio acoustic wind profiler data in the SJV, reported by Bianco et al. (2011), could be used to estimate entrainment rates by simply measuring the boundary layer growth during the midday.

This idea is explored in Fig. 6, where we show the monthly average ABL growth rates observed year round by NOAA's wind profiler network operated across California's Central Valley during 2008 (Bianco et al., 2011). Additionally, the monthly average subsidence at boundary layer height is shown as captured in the NARR data set. Assuming that the advection term does not dominate at any of the sites in a long term average (other than at Lost Hills where it is possibly counterbalanced by the convergent uplift), we can get a sense of the general entrainment characteristics across the Central Valley throughout the year. For example, there appears to be stronger entrainment at lower latitudes in the valley (~3 cm s$^{-1}$ annual peaks in the Sacramento vs. ~4 cm s$^{-1}$ peaks in the San Joaquin Valley), possibly due to greater shortwave forcing or generally weaker stratification in the lower FT. It further seems that at most sites there is a definite peak in entrainment during the Spring but also a secondary maximum in the Autumn with a minimum during the mid–summer. This corresponds to the lowest ABL depths observed in the middle of summer as discussed by Bianco et al. (2011). In their analysis the authors suggest that the lower inversion heights of mid-summer are due to greater cold air advection through the delta and/or possibly the peak in irrigation in the heavily agriculturally controlled landscape of the Central Valley. Both effects serve to cool the ABL thereby increasing lower tropospheric stability (LTS) and suppressing entrainment. The lower right panel in Fig. 6 shows the LTS, as measured by the difference in potential temperatures between 750–900 hPa. The LTS minima in Spring and Autumn appear to coincide approximately with the peaks in entrainment more or less across the entire Central Valley.

### 3.2 Other Budget Residuals

Once the entrainment rate has been calculated for each flight it can be used to close the other scalar budget equations and calculate any single residual term, assuming all the others have been characterized. The time derivative and gradient terms were all calculated by applying a simple multi-linear regression on all flight data collected below the (time varying) ABL height as described in Conley et al. (2010).

#### 3.2.1 Ozone Photochemical production

Figure 7 illustrates the distinction between ABL and FT air and the importance of entrainment mixing on an ozone exceedance day. The potential temperature and specific humidity on the left graph show the surface heating and nearly well–mixed ABL capped by the stable inversion with dry, warm air aloft. The right hand graph shows the enhanced NO$_2$ and O$_3$ within the ABL





during the day because of the surface emissions of $NO_x$ and the photochemical production of $O_3$ from those emissions in conjunction with volatile organic compounds (VOC). The ABL top, $z_i$, is indicated by the dashed line near 850 m. Given the jumps in $O_3$ and $NO_2$ evident at that height, and the estimated mean entrainment velocity for the entire flight, 5.1 cm s$^{-1}$ (Table 1), the effect of entrainment dilution alone is causing a drop in surface $O_3$ and $NO_2$ concentrations by 4.0 and 0.32 ppb h$^{-1}$,

respectively, demonstrating how important entrainment can be for understanding the temporal evolution of air pollutants measured near the surface. The consequences of horizontal advection can be seen in Fig. 8, which shows the spatial distribution of $O_3$ and $NO_2$ during the same day, 14 August 2013. Because $O_3$ rises steadily throughout the flight, all the data is corrected to a common time (13:30) by the observed mean temporal trend of 2.4 ppb h$^{-1}$. The spatial pattern shows a strong negative $O_3$ advection of -2.5 ppb h$^{-1}$ into the Arvin area, but a countervailing positive $NO_2$ advection. Thus while consideration of the $O_3$

budget requires taking into account this inflow of lower $O_3$, the selfsame flow carries with it abundant precursors that boost the in-situ $O_3$ production near Arvin, the term that we infer through closure of the overall budget. This distribution of higher $O_3$ around Arvin was not observed on every day, but was more common on ozone exceedance days. Because we only measured the $NO_2$ distribution once, it is more difficult to generalize, but the local maximum of $NO_2$ near Bakersfield has been reported elsewhere and is evident in seasonal satellite averages reported in Russell et al. (2010) and Pusede & Cohen (2012).

In addition to applying our derived entrainment rates to close the $O_3$ budget (Eq. (4), results from which are presented in Table 2) we estimated the dry deposition term using a deposition velocity of 0.5 cm s$^{-1}$, with an estimated uncertainty of ± 0.25 cm s$^{-1}$, based on values reported in the literature for similar environments (Padro, 1996; Macpherson et al., 1995; Pio et al., 2000). The deposition term is the product of the deposition velocity and average ABL concentration divided by the ABL height. Dry deposition velocities are often reported with respect to a 10 m measurement, and although the lowest safe flight altitude is 150 m

and we therefore do not have $O_3$ measurements at 10 m (aside from take offs and landings), the vertical gradients of $O_3$ tend to be no more than about 1 ppb per 100 m (Fig. 7), so we consider the uncertainty in the 10 m concentration to be ~ 2 ppb (3–4 % the mean $O_3$), and insignificant compared to the uncertainty in the deposition velocity of 50%. Ozone photochemical production (P) was estimated to be between 4.1 and 14.2 ppb h$^{-1}$ in summer and 2.1–3.9 ppb h$^{-1}$ in the winter. Comparisons between the winter and summer data sets are relevant. Although differences between the two sites could, in principle, arise due to varying local

sources between Fresno and Bakersfield, the photochemical production is expected to be much lower in the winter with reduced actinic radiation fluxes. Note that there is a near tripling of the photochemical production between the two seasons, in winter the average is 2.8 ppb h$^{-1}$ and in summer 8.2 ppb h$^{-1}$. $O_3$ production in the southern SJV, during the warm season, is believed to be $NO_x$-limited for most conditions except for weekdays (higher $NO_x$ on average) when temperatures exceed 29° C, as proposed by Pusede et al. (2014), who investigated various factors in the production of ozone in the SJV. All of those conditions were met for

the flights in the ArvinO3 study, and with the continued decrease in $NO_x$ expected from the 7 year trend of -32% in Bakersfield reported by Russell et al. (2012) based on OMI satellite measurements of column $NO_2$, the conditions are only becoming more and more frequently $NO_x$–limited. A VOC:$NO_x$ ratio proxy was derived from the airborne measurements of methane minus the global      background      methane      from      NOAA's      Global      Greenhouse      Gas      Reference      Network (http://www.esrl.noaa.gov/gmd/ccgg/trends_ch4/) divided by the CARB surface air quality monitoring network $NO_x$

concentrations measured during the flight hours. Although the VOC makeup of the SJV is fairly complex due to the preponderance of dairy farms and natural gas production, both of these source types are strong methane emitters (Gentner et al., 2014), so we consider observed methane to be a decent proxy for the overall abundance of non-methane VOCs. Figure 9 shows



that the inferred $O_3$ production rates from both studies (Table 2) decrease with increasing VOC:NOx ratio proxy indicating that the SJV is mostly under $NO_x$–limited conditions.

Using a simplified box model constrained by observations of $NO_x$, $HO_x$, and VOC reactivity, Pusede et al. (2014) estimate $O_3$ production rates ranging from 10–26 ppb h$^{-1}$ at the Bakersfield CalNex supersite. This is approximately double the rates reported in this study using the budgeting technique, 4–14 ppb h$^{-1}$ (Table 2). The results reported by Pusede et al. (2014) are not net, but only the sum of the $O_3$ photochemical production channels. However, Pusede et al. (2014) estimate that the $O_3$ photochemical loss rates rarely exceed ~1.5 ppb h$^{-1}$, and we thus assume that this can only be a small part of the difference between our estimates and theirs. A much more significant difference is likely because of the fact that Pusede et al. (2014) use measurements made inside the metropolitan area of Bakersfield, while the flight data represents a region of about 4600 km$^2$ in which most of the land use is agricultural. Therefore we expect the regional $O_3$ production to be smaller because it incorporates land outside of the urban centre where the $NO_x$ is likely to be considerably lower on average (Pusede & Cohen, 2012).

Baidar et al. (2013) performed a budget study based on a research flight conducted on June 15, 2010 in and around Bakersfield. Amongst the objectives of the study was the determination of an emission rate for $NO_x$ and $O_x$ ($O_3$ + $NO_2$) production rates from the urban area. They attempted a similar scalar budget approach using flight data obtained by remote sensing instruments (3 different lidar systems) aboard the NOAA Twin Otter, obtaining a range of $O_3$ production rates from 2.9–6.6 ppb h$^{-1}$ with an area weighted average of 4.0 ppb h$^{-1}$. Within their volume of interest, they assumed the time rate of change of $NO_x$ and $O_x$ were zero, assuming that the horizontal flux divergence alone determines the source strength for their region. Aside from temporal changes (storage terms), they further neglected entrainment and dry deposition fluxes of these constituents. From their Fig. 5 indicating the diurnal signal of $NO_x$ and $O_x$ taken from the Bakersfield CARB monitoring station, we estimate a 2.2 ppb h$^{-1}$ change in $O_3$ during their measurement time. In addition, they estimate a potential error of not including vertical mixing, or entrainment, to be less than 2%. Their estimate of the entrainment rate is on the low end of our range, at 1.2 cm s$^{-1}$, but when calculating their entrainment flux they use a delta $O_x$ of about -4 ppb. Using our average observed jump across the ABL top of -13.4 ppb, an average entrainment velocity of 3.0 cm s$^{-1}$, and an average boundary layer height of 1000 m, along with a dry deposition velocity of 0.5 cm s$^{-1}$, the vertical terms give rise to a loss rate of approximately 2.6 ppb h$^{-1}$. This could easily explain the difference between their average of 4.0 ppb h$^{-1}$ and our average of 8 ppb h$^{-1}$. But the comparison is imperfect because the ArvinO3 study specifically targeted ozone exceedance events (albeit only capturing 4 NAAQS and 6 California state exceedance days out of 11 flight days). During the day of the Baidar et al. (2013) study the $O_3$ peaked at only ~65 ppb based on the CARB surface monitoring network. Nevertheless, the comparison further points to the importance of treating all the budget terms in estimating net photochemical $O_3$ production. In our study the contribution to the $O_3$ budget from entrainment dilution is typically the same magnitude as the observed rise in $O_3$, and the latter alone only constitutes one-third of the total net production.

### 3.2.2   Methane Emission

For a scalar such as methane partaking in extremely slow chemistry (with a photochemical lifetime of about a decade), the budget equation can be easily solved for the surface emission rate:

$$F_s = \left( U \frac{\partial CH_4}{\partial x} + \frac{\partial CH_4}{\partial t} \right) z_i + F_{ent} \tag{7}$$

where the advection and temporal trend terms are observed directly by the aircraft, and $F_{ent}$, the entrainment flux, is estimated using the parameterization of Eq. 5 based on the observed jump in $CH_4$ across the ABL top and the entrainment velocity derived





from the ABL height budget. Regional methane emissions from the DISCOVER–AQ campaign near Fresno were estimated to be $100 \pm 100$ Gg yr$^{-1}$, and from the Arvin-Bakersfield region they were estimated to be $170 \pm 125$ Gg yr$^{-1}$ when averaged over each respective flight campaign. The second numbers reported above represent the estimated standard deviation of the mean value representing the spread in the measurements across the different days of each campaign, not the estimated error in the measurements themselves. To obtain our in-situ emission estimate we multiplied our regionally averaged surface methane emission by the approximate horizontal area encompassed by the series of flights. For flight region one we estimated the horizontal area to be $9.5 \times 10^8$ m$^2$ because the flight pattern was simply across valley and the horizontal winds were light so there was little need to probe the direction of the mean advection. Flight region two covered a much larger area of $3.5 \times 10^9$ m$^2$ because the experiment specifically targeted a careful mapping of the up-valley advection term in the $O_3$ budget. In a recent work by Kort et al. (2014) using the Scanning Imaging Absorption Spectrometer for Atmospheric Chartography (SCIAMACHY) instrument from 2003–2009 the column-averaged $CH_4$ mole fractions over the U.S. are used to estimate surface emissions. Although the thrust of that study was the 'hot spot' observed over the four corners region of New Mexico, it is interesting to note that the second largest spot (their Fig. 1) that emerges in the satellite climatology is located in the southern San Joaquin Valley of California. Using the California Greenhouse Gas Emission Measurement (CALGEM; http://calgem.lbl.gov/prior_emission.html) inventory we estimated the emissions from each sector for both flight regions. The emission estimates have been scaled to the 2013 total $CH_4$ emission estimate for California of 41.1 TgCO2eq provided by CARB. Inventory emissions from flight region one was found to be a total of 27.7 Gg yr$^{-1}$ and from region two 71.1 Gg yr$^{-1}$. Comparing these to the in-situ estimates of this study we find our estimates to be 3.6 and 2.4 times greater than the scaled CALGEM inventory estimates, respectively. According to the breakdown in sources found in the CALGEM database we estimated the fractional coverage of each source type for the two experiments. The first region sampled in winter near Fresno for the DISCOVER–AQ project was found to bear 54% fossil fuel related sources, with the majority of the balance coming from dairies (25%) and other livestock (9%) and landfills (11%). Flight region two flown during the summertime around Bakersfield was more dominated by dairies (73%), with most of the rest fossil related (17%). The difference in make-up of the two regions is broadly consistent with the finding expounded by Miller et al. (2013) that ruminant sources of methane appear to be approximately twice as large as current inventories hold, while fossil fuel sources are nearly six times larger than the present inventories indicate. This could account for the greater discrepancy found in the DISCOVER–AQ data where observed emissions are more heavily influenced by sources associated with fossil fuels.

To further examine the observed variability of the methane emissions in the southern SJV, where the sources are predominantly from dairies and thus derive from enteric and manure methanogenesis, the temperature dependence is presented in Fig. 10 in an Arrhenius type plot. In general, the temperature response of microbial activity (ultimately the source of methane emission associated with livestock) is often quantified by an Arrhenius equation: i.e., $\text{rate} = A \cdot \exp(-Ea/RT)$, where $A$ is a pre-exponential factor, $E_a$ is the activation energy, $R$ is the universal gas constant (8.314 J mol$^{-1}$ K$^{-1}$), and $T$ is the absolute temperature. Figure 10 shows the natural log of our estimates of methane emissions, at temperatures below the optimum (peak methane production occurs in the mesophillic range of 30–37°C). The results of Elsgaard et al. (2016) indicate a peak in methane production near 38°C in cattle slurries. In order to compare most appropriately, we removed the $CH_4$ emission rate estimate of the flight of 9 June 2014 when the air temperatures exceeded 39°C, and we set the emission estimate to 0 (from -20 Gg yr$^{-1}$, within the method's uncertainty) for the 30 September flight, which was the coldest day of the experiment (afternoon average surface temperature in Bakersfield of 25.7°C). The resultant data in Fig. 10 shows signs of an Arrhenius type behaviour in the



dominant methane sources in the southern end of the SJV, and moreover the activation energy, $E_a$, derived from the fit is 76 kJ mol$^{-1}$ is very similar to that found by Elsgaard et al. (2016) of 81 kJ mol$^{-1}$. The correlation coefficient for the linear fits does not change significantly when the flight data from the two dates mentioned above are included ($r^2$ of 0.54 instead of 0.58).

### 3.2.3  Surface Latent Heat Flux

Rearrangement of the water budget relationship Eq. (6), in a fashion similar to that of methane, leads to the ready estimation of surface latent heat fluxes for each campaign. The average for summer flights around Bakersfield was 284 W m$^{-2}$ and for winter outside of Fresno it was 90 W m$^{-2}$. Comparing these values to reference evapotranspiration estimated by the CIMIS network (515 and 160 W m$^{-2}$, respectively) we find that both experiments predict virtually identical fractions, 55%, occurring across the regions. This is likely the result of mixed land uses dominated by agriculture with interspersed fallow and actively growing plots. As expected the latent heat fluxes were observed to be lower in winter as the solar radiation is smaller and crop demand for water is reduced, but in both seasons it was found to be dramatically larger than the surface latent heat fluxes used in the NARR reanalysis data. This result, which indicates the lack of accurate irrigation information in the NARR land surface model, is significant because it is most likely the reason why the reanalysis data has boundary layers that are very much higher than observed, therefore this data should be used with caution by the community.

### 4  Error Analysis

All gradient terms from the scalar budgets were calculated using a multiple linear regression and the error in them was taken as a standard error of the estimate. Omega values taken from NARR reanalysis were assumed to be ± 0.05 Pa s$^{-1}$ (~ 0.5 cm s$^{-1}$) as a best guess estimate. Other terms were also estimated, including; the scalar delta terms for the jump across the entrainment zone based on observing their vertical profiles, or taken from instrument specifications. All terms were combined and added in quadrature to obtain an error for compound terms.

### 5  Conclusions

In situ measurement via targeted aircraft campaigns can help us understand key factors in boundary layer dynamics, including entrainment. It is propitious when it comes to probing complex mesoscale features, i.e. areas influenced by mountain–valley dynamics. A better understanding of entrainment is integral to understanding air quality on the ground, and it has potential applications in quantifying the significance of trans–boundary contributions. The simple, yet novel scalar budgeting technique outlined here is invaluable to boundary layer studies, and it is easy to apply with sufficient probing of the ABL for any given region. From our analysis of the inversion height budget, the boundary layer height advection balances the mean upward vertical wind forced by orographic convergence at the southern end of the SJV. This balance permits the measurement of entrainment by simply measuring the change in ABL height throughout the daytime. The NOAA RASS sounders would suffice in this region to make regular measurements of entrainment, and analysis of data reported by Bianco et al. (2011) from 2008 shows bimodal peaks in entrainment in early spring (March) and late summer (August) at Lost Hills approximately 40 km northwest of Bakersfield. Similar bimodal peaks in entrainment were found during spring and autumn for sites throughout California's Central Valley, and may be due to the minima LTS in those transition seasons.



Applying the entrainment results of the budgeting of ABL height to the other scalars then leads to significant insights into their sources and controlling variables. It was found that entrainment dilution and dry deposition of $O_3$ are comparable in magnitude (but opposite in sign) to the observed time rate of change, which itself is only one-third of the net photochemical production during the $O_3$ season in the Bakersfield/Arvin area. While advection of $O_3$ into the town of Arvin is consistently observed to be

negative (lower $O_3$ air being brought in by the up–valley flow), a steady advection of high $NO_x$ upstream seems to keep the in situ production elevated in the Arvin area. Moreover, a proxy for VOC to $NO_x$ ratio was used from the airborne methane and the surface air quality network $NO_x$ to show that $O_3$ production is $NO_x$–limited in the southern SJV in summer and mid–SJV in the winter. The methane budgets revealed stronger sources in the SJV than those in the CALGEM database, with a greater disparity in the wintertime near Fresno, where there is a greater fraction of methane from petroleum related sources. And finally the water

vapour budget showed that the evapotranspiration in these regions are approximately 55% of their reference values (with respect to well watered and groomed grass) according to the CIMIS network in both seasons. These evapotranspiration rates are much larger than contained in the NARR data set, which does not appear to include realistic irrigation in its land surface module, and this will be a source of significant overestimation of boundary layer heights throughout the year in the Central Valley.

### Acknowledgements

The ArvinO3 study was made possible by backing from the San Joaquin Valley Air Pollution Control District. We would especially like to thank David Lighthall, may he rest in peace, for his enthusiastic support, many fruitful discussions, and for his friendship. Flight time to participate in NASA's DISCOVER-AQ was provided by the Bay Area Air Quality Management District, and we thank the former's Jim Crawford and the latter's Saffet Tanrikulu for making it happen. We are very indebted to Irina Djalalova, Laura Bianco, and James Wilczak for providing us with their RASS boundary layer height data from across the

Central Valley. We further thank Doug Baer and his colleagues at Los Gatos Research for the generous loan of their $NO_2$ spectrometer, and Marc Fischer for freely sharing his CALGEM methane source inventory.

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



| | dzi/dt (cm s⁻¹) | Sfc-P Tend (cm s⁻¹) | Omega (cm s⁻¹) | W (cm s⁻¹) | $z_i$ advection (cm s⁻¹) | $w_e$ (cm s⁻¹) |
|---|---|---|---|---|---|---|
| 1/16/13 | 1.13(0.29) | -0.16 (0.01) | 0.8(0.5) | -1.0(0.5) | 0.08(0.17) | 2.08(0.79) |
| 1/18/13 | 1.14(0.60) | -0.16(0.01) | 0.4(0.5) | -0.6(0.5) | 0.05(0.47) | 1.69(1.10) |
| 1/20/13 | 0.56(0.63) | -0.16(0.01) | 0.8(0.5) | -1.0(0.5) | 0.04(0.25) | 1.56(1.13) |
| 1/21/13 | 0.72(0.22) | -0.11(0.01) | -0.9(0.5) | 0.9(0.5) | 0.07(0.08) | -0.21(0.72) |
| 1/22/13 | 1.85(0.49) | -0.16(0.01) | -0.1(0.5) | 1.1(0.5) | -0.21(1.14) | 0.96(0.99) |
| 1/30/13 | 3.57(0.58) | -0.05(0.01) | -1.7(0.5) | 1.6(0.5) | -0.44(0.14) | 2.39(1.08) |
| 2/4/13 | 1.38(0.60) | -0.21(0.01) | 0.5(0.5) | -0.8(0.5) | 0.06(0.19) | 2.13(1.10) |
| Averages | 1.48(0.49) | -0.14(0.01) | -0.0(0.5) | -0.0(0.5) | -0.05(0.35) | 1.51(0.99) |
| | | | | | | |
| 6/26/13 | 1.23(0.39) | -0.12(0.01) | 0.3(0.5) | -0.4(0.5) | 0.75(1.70) | 0.86(0.89) |
| 6/27/13 | 3.79(0.72) | -0.12(0.01) | -2.7(0.5) | 2.5(0.5) | -0.20(1.75) | 1.51(0.72) |
| 6/28/13 | 4.28(1.17) | -0.18(0.01) | -1.5(0.5) | 1.3(0.5) | -1.67(4.44) | 4.69(1.67) |
| 8/13/13 | 4.11(0.35) | -0.17(0.01) | -0.8(0.5) | 0.6(0.5) | -1.47(1.96) | 4.98(0.85) |
| 8/14/13 | 2.42(0.79) | -0.12(0.01) | -1.7(0.5) | 1.5(0.5) | -4.24(3.36) | 5.14(1.29) |
| 8/15/13 | 2.54(0.75) | -0.11(0.01) | -2.6(0.5) | 2.5(0.5) | -1.03(0.96) | 1.10(1.25) |
| 9/28/13 | 1.20(0.02) | -0.17(0.01) | 0.1(0.5) | -0.3(0.5) | 0.00(0.03) | 1.45(0.52) |
| 9/29/13 | 2.75(0.47) | -0.17(0.01) | -1.9(0.5) | 1.7(0.5) | -0.02(0.12) | 1.05(0.97) |
| 9/30/13 | 2.95(0.46) | -0.17(0.01) | -1.1(0.5) | 0.9(0.5) | 0.81(3.34) | 1.23(0.96) |
| 6/8/14 | 4.95(0.85) | -0.12(0.01) | -1.4(0.5) | 1.2(0.5) | -0.70(0.49) | 4.41(1.35) |
| 6/9/14 | 2.83(1.00) | -0.21(0.01) | -1.5(0.5) | 1.2(0.5) | -4.84(7.12) | 6.52(1.50) |
| Averages | 3.00(0.63) | -0.15(0.01) | -1.3(0.5) | 1.2(0.5) | -1.15(2.30) | 2.99(1.09) |

**Table 1: Entrainment rate table for both studies (DISCOVER-AQ in the winter of 2013 near Fresno, and the Arvin O3 study in the summers of 2013/14 near Bakersfield), $z_i$ budget terms, the surface pressure tendency, omega at the level nearest the observed ABL height (from the NARR data set), and subsidence. The surface pressure tendency and omega values are corrected to linear speeds via the hydrostatic approximation for the sake of comparison. Values in parentheses represent estimates of each term's 1 $\sigma$ uncertainty.**





| | dO3/dt (ppb h⁻¹) | O3 Advec (ppb h⁻¹) | Dep O3 (ppb h⁻¹) | Ent O3 (ppb h⁻¹) | Δ O3 (ppb) | Photo Prod. (ppb h⁻¹) | Avg O3 ABL (ppb) |
|---|---|---|---|---|---|---|---|
| 1/16/13 | 0.70(0.002) | -0.02(0.01) | -0.91(0.59) | -0.75(0.32) | -5.0(1.0) | 2.34(0.92) | 38.4(1.0) |
| 1/18/13 | 1.09(0.003) | -0.05(0.05) | -1.02(0.75) | -0.47(0.28) | -4.0(1.0) | 2.63(1.08) | 41.5(1.0) |
| 1/20/13 | 0.72(0.002) | 0.25(0.09) | -1.04(0.76) | -0.60(0.34) | -6.0(1.0) | 2.10(1.18) | 46.0(1.0) |
| 1/21/13 | 1.15(0.003) | 0.25(0.05) | -1.31(4.08) | 0.08(0.37) | -8.0(1.0) | 2.13(4.50) | 52.2(1.0) |
| 1/22/13 | 0.97(0.005) | -0.37(0.03) | -1.25(1.19) | -0.21(0.27) | -5.0(1.0) | 2.87(1.50) | 52.9(1.0) |
| 1/30/13 | 2.21(0.001) | 0.11(0.02) | -0.70(0.44) | -0.06(0.26) | -5.0(1.0) | 3.39(0.72) | 40.4(1.0) |
| 2/4/13 | 2.19(0.003) | 0.02(0.03) | -0.92(0.59) | -0.84(0.36) | -7.0(1.0) | 3.92(0.98) | 46.0(1.0) |
| Averages | 1.29(0.003) | 0.03(0.04) | -1.02(1.20) | -0.41(0.31) | -5.7(1.0) | 2.77(1.56) | 45.3(1.0) |
| | | | | | | | |
| 6/26/13 | 2.48(0.002) | -2.19(0.16) | -0.82(0.75) | -6.55(0.73) | -40.0(1.0) | 11.89(1.65) | 60.0(1.0) |
| 6/27/13 | 3.72(0.002) | -1.00(0.12) | -0.81(0.68) | -1.79(0.57) | -20.0(1.0) | 7.33(1.37) | 56.8(1.0) |
| 6/28/13 | 3.12(0.002) | -1.54(0.13) | -1.02(0.60) | -1.20(0.66) | -16.0(1.0) | 6.89(1.39) | 72.9(1.0) |
| 8/13/13 | 1.18(0.001) | -9.93(0.22) | -1.21(0.64) | -1.91(0.28) | -8.0(1.0) | 14.23(1.14) | 72.2(1.0) |
| 8/14/13 | 2.35(0.002) | -2.48(0.23) | -1.28(0.71) | -0.96(0.40) | -10.0(1.0) | 7.06(1.34) | 79.2(1.0) |
| 8/15/13 | 2.68(0.002) | -4.98(0.30) | -1.50(1.55) | -2.34(0.50) | -13.0(1.0) | 11.51(2.35) | 77.2(1.0) |
| 9/28/13 | 3.71(0.001) | -0.66(0.06) | -1.31(0.82) | -2.29(0.24) | -10.0(1.0) | 7.92(1.12) | 61.3(1.0) |
| 9/29/13 | 2.14(0.001) | 0.23(0.03) | -1.65(1.57) | -1.77(0.26) | -6.0(1.0) | 5.34(1.86) | 66.2(1.0) |
| 9/30/13 | 4.77(0.001) | -0.29(0.03) | -0.70(0.60) | -1.27(0.17) | -5.0(1.0) | 7.04(0.80) | 36.7(1.0) |
| 6/8/14 | 2.73(0.002) | 1.56(0.11) | -1.21(0.69) | -1.77(0.41) | -10.0(1.0) | 4.15(1.21) | 76.9(1.0) |
| 6/9/14 | 1.65(0.002) | -0.34(0.07) | -1.49(0.81) | -2.77(0.51) | -10.0(1.0) | 6.28(1.39) | 92.2(1.0) |
| Averages | 2.78(0.002) | -1.97(0.13) | -1.18(0.86) | -2.24(0.43) | -13.5(1.0) | 8.15(1.42) | 68.3(1.0) |

**Table 2: Ozone Budgets for the DISCOVER–AQ mission (top) and the ArvinO3 mission (below). Values in parentheses represent estimates of 1 σ uncertainties in each measurement.**





| | dCH4/dt (ppmv h⁻¹) | CH4 advec (ppmv h⁻¹) | CH4 Ent Flux (ppmv h⁻¹) | Δ CH4 (ppmv) | CH4 Prod (Ggrams yr⁻¹) | Avg CH4 ABL (ppmv) |
|---|---|---|---|---|---|---|
| 1/16/13 | 0.0231(0.006) | 0.000(0.0004) | -0.056(0.046) | -0.40(0.05) | 237.99(142.53) | 2.13(0.002) |
| 1/18/13 | 0.0260(0.0013) | 0.004(0.0016) | -0.024(0.024) | -0.20(0.05) | 131.43(80.45) | 2.23(0.002) |
| 1/20/13 | -0.0029(0.0009) | 0.008(0.0014) | -0.030(0.031) | -0.30(0.05) | 58.37(103.47) | 2.21(0.002) |
| 1/21/13 | -0.0100(0.009) | 0.006(0.0043) | 0.00(0.021) | -0.40(0.05) | -57.99(74.14) | 2.31(0.002) |
| 1/22/13 | -0.1390(0.0018) | -0.01(0.01) | -0.019(0.024) | -0.30(0.05) | 53.31(108.15) | 2.29(0.002) |
| 1/30/13 | 0.0254(0.0008) | -0.001(0.0013) | -0.022(0.021) | -0.20(0.05) | 204.96(97.10) | 2.02(0.002) |
| 2/4/13 | -0.0003(0.0005) | 0.004(0.0021) | -0.024(0.021) | -0.20(0.05) | 70.77(87.4) | 2.07(0.002) |
| Averages | -0.011(0.001) | 0.001(0.003) | -0.024(0.022) | -0.29(0.05) | 99.83(99.03) | 2.18(0.002) |
| | | | | | | |
| 6/26/13 | 0.0004(0.0001) | -0.006(0.0091) | -0.004(0.005) | -0.16(0.05) | 254.80(96.26) | 1.90(0.002) |
| 6/27/13 | -0.0014(0.0001) | -0.005(0.0047) | -0.007(0.008) | -0.15(0.05) | 261.00(78.84) | 1.87(0.002) |
| 6/28/13 | -0.0140(0.0004) | -0.013(0.0049) | -0.013(0.012) | -0.10(0.05) | 282.00(116.24) | 1.92(0.002) |
| 8/13/13 | | | | | | |
| 8/14/13 | | | | | | |
| 8/15/13 | | | | | | |
| 9/28/13 | -0.0105(0.0004) | -0.005(0.0086) | -0.013(0.011) | -0.21(0.05) | 117.00(88.78) | 2.08(0.002) |
| 9/29/13 | -0.0019(0.0003) | 0.01(0.0117) | -0.010(0.012) | -0.19(0.05) | 16.00(90.65) | 2.07(0.002) |
| 9/30/13 | -0.0066(0.0003) | 0.005(0.0143) | -0.011(0.010) | -0.22(0.05) | -25.00(121.93) | 1.94(0.002) |
| 6/8/14 | -0.0250(0.0006) | -0.011(0.0022) | -0.021(0.014) | -0.15(0.05) | 150.50(96.52) | 2.08(0.002) |
| 6/9/14 | -0.0150(0.0004) | -0.004(0.003) | -0.025(0.018) | -0.12(0.05) | 300.00(114.88) | 2.08(0.002) |
| Averages | -0.0092(0.0003) | -0.004(0.007) | 0.013(0.011) | -0.16(0.05) | 169.54(100.51) | 1.99(0.002) |

**Table 3: Methane Budgets for the DISCOVER–AQ mission (top) and the ArvinO3 mission (below). Values in parentheses represent estimates of 1 σ uncertainties in each measurement.**





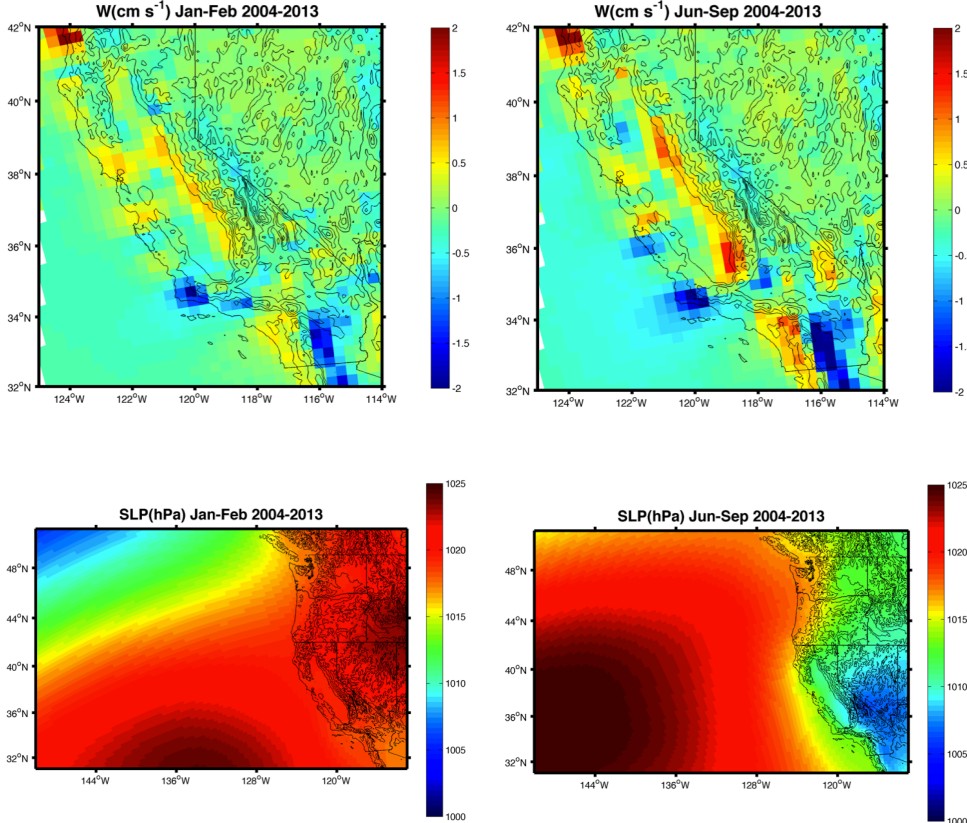

**Figure 1: W (vertical velocity), converted from omega (pressure velocity), at the 900 hPa level and mean sea level surface pressure. Plotted for two intervals Jan–Feb and June–Sept. for 10 years from 2004–2013. The months chosen for the two separate plots represent the time frame of the flights.**



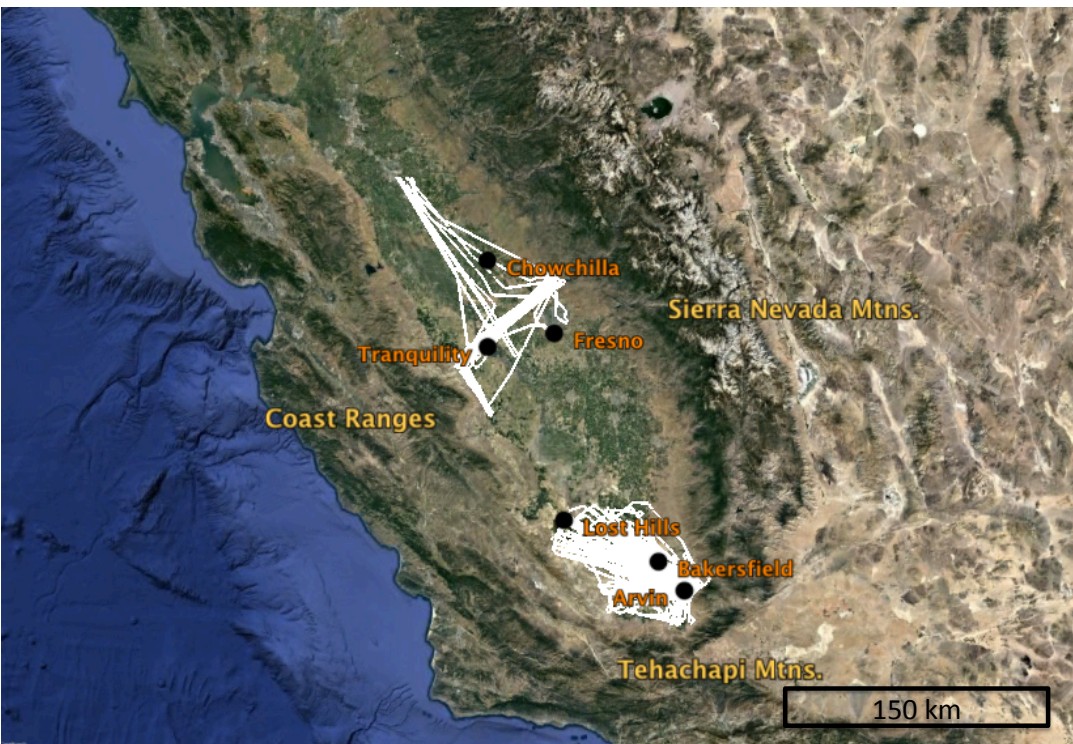

**Figure 2: Flight paths of all data observed in the ABL for the two projects of this study: DISCOVER–AQ near Fresno from Jan/Feb 2013, and the Arvin O3 project sampling from June–Sept over two summers and carefully mapping the inflow region upwind of Bakersfield and Arvin.**





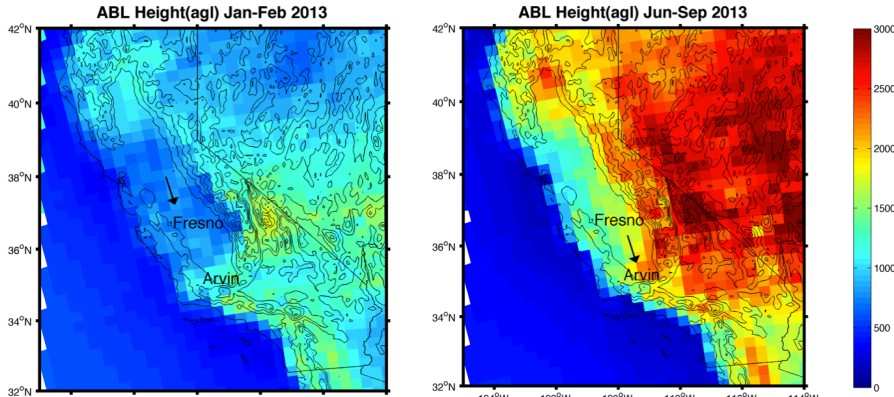

5   **Figure 3:  The average spatial pattern of boundary layer heights from the NCEP/NARR data set for (left) the winter period, and (right) the summer period of this study. Wind vectors represent the mean in situ winds measured by the aircraft near the ABL top.**





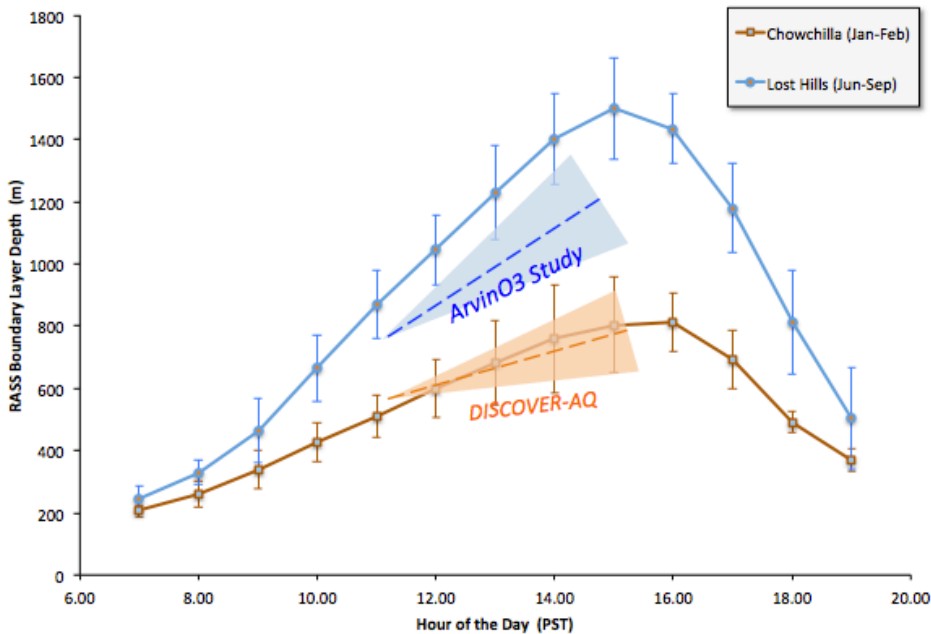

**Figure 4: Diurnal boundary layer development as observed during the two experiments presented here, and the average data from the corresponding months and locations presented in Bianco et al. (2011).**




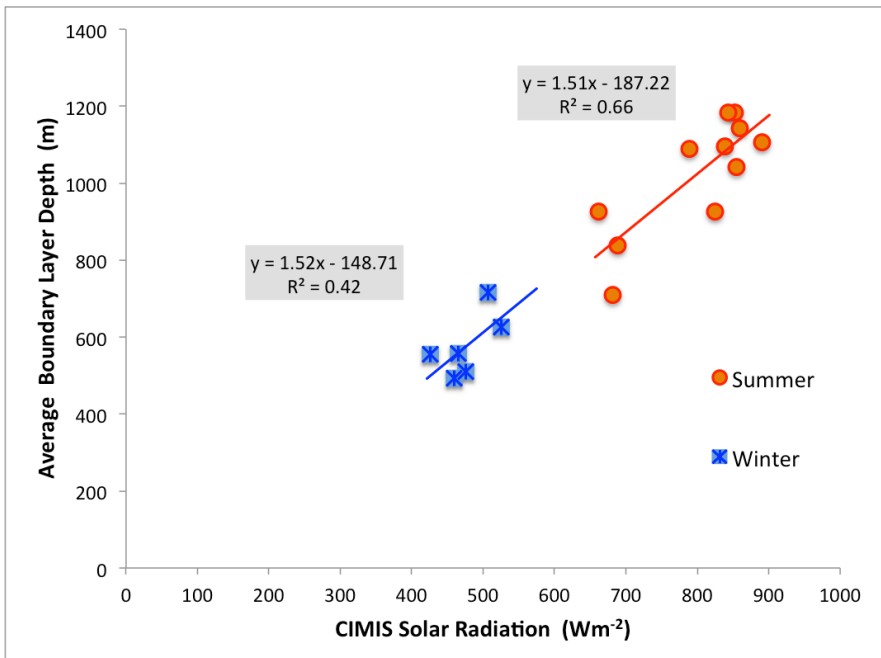

**Figure 5:** Flight averaged boundary layer depths as a function of the surface downwelling solar radiation as measured by the CIMIS station pyranometer in the flight regions near Fresno in winter, and Bakersfield in summer.



**Figure 6:** Observed monthly average ABL growth rates (dzi/dt, green lines) from the RASS network across the Central Valley described in Bianco et al. (2011) from the entire year of 2008. The mean vertical wind from the NCEP/NARR data set is included (blue lines) to yield estimates of entrainment (red lines). The lower right panel depicts the lower tropospheric stability (LTS) defined from the reanalysis data as the difference in potential temperatures at 750 and 900 hPa levels.





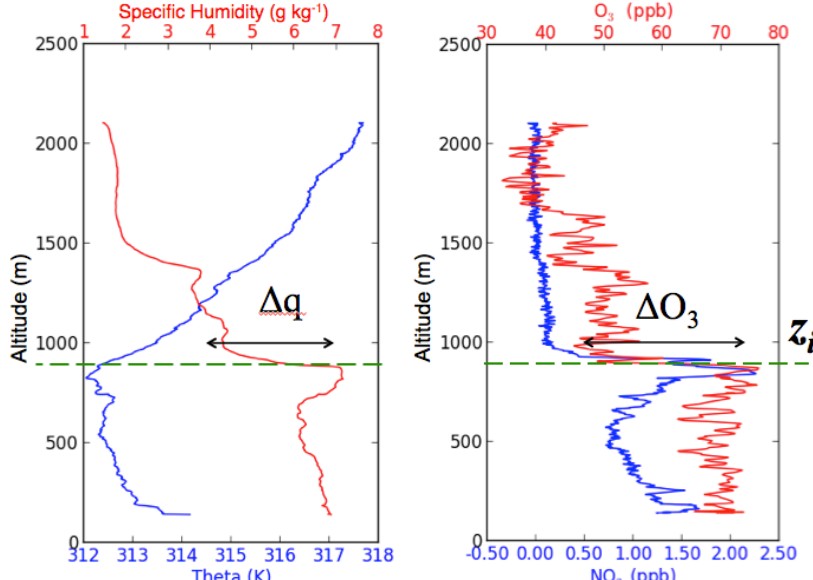

**Figure 7:** **Example of vertical profiles of potentital temperature (theta) and specific humidity (q) on the left, and ozone and NO₂ observed on the right during the flight on 14 August 2013 near Bakersfield, CA. $Z_i$ is the estimated height of the ABL determined by the scalar jump (Δq and ΔO₃ shown here) across the entrainment zone.**





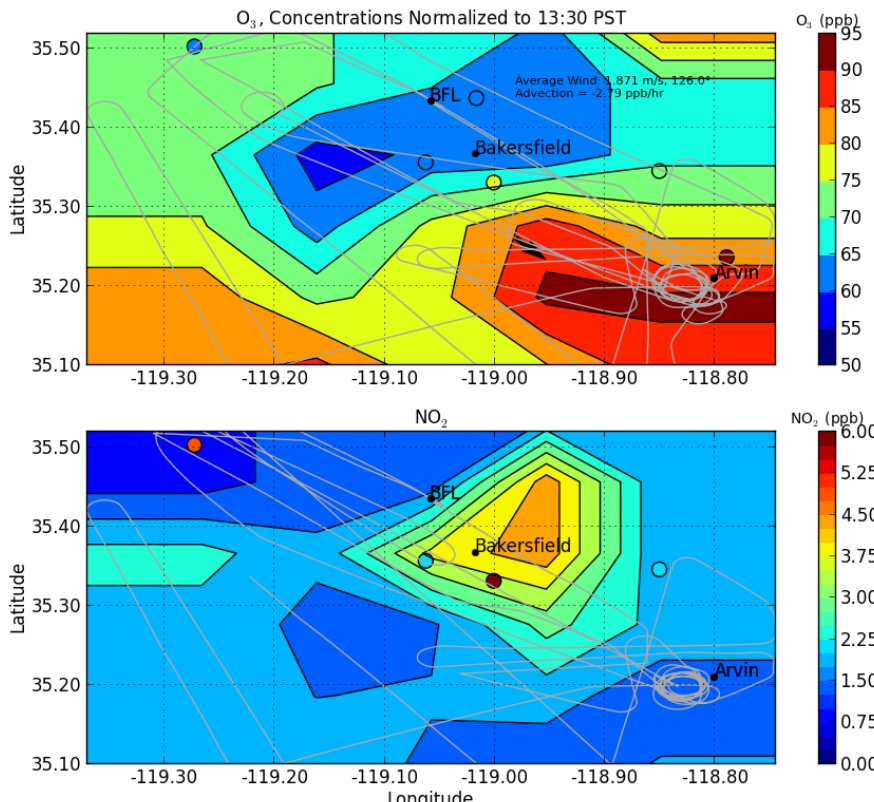

**Figure 8:** Horizontal patterns of O₃ (top) and NO₂ (bottom) during an ozone exceedance episode near Bakersfield on 14 August 2013. The gray lines represent the flight tracks, and the coloured circles represent the surface network observations. Because of the continual trend in ozone throughout the flight, the values in the top figure are all corrected to a reference time of 13:30 PST. The black arrow in the top figure represents the vector average wind observed in the ABL during that sortie showing a strong negative advection of ozone and a large positive advection of NO₂ into the Arvin region to the south. BFL is the Meadows Field Airport on the north end of urban Bakersfield.



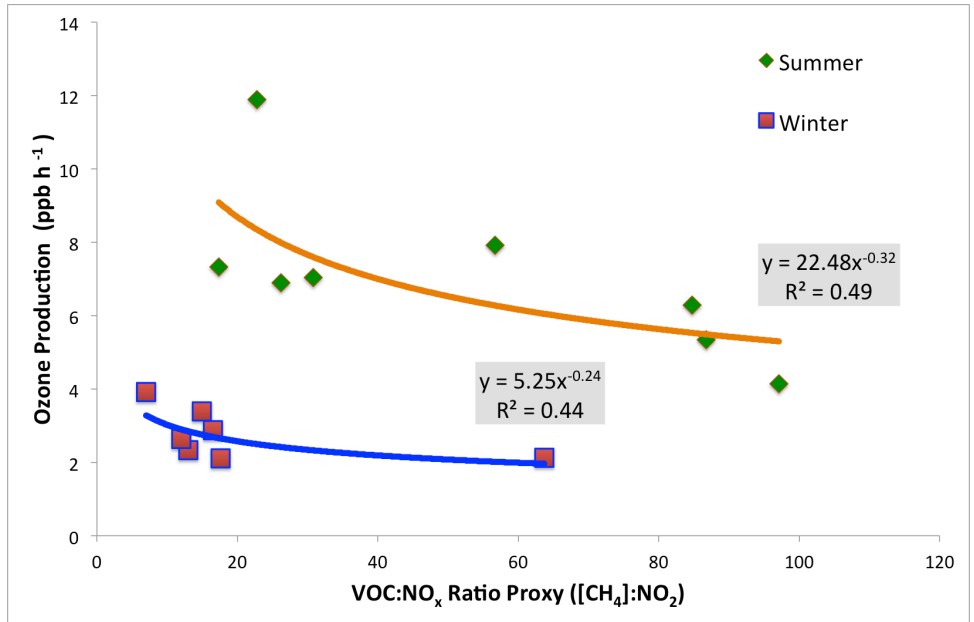

**Figure 9: Plot of measured ozone production from the DISCOVER-AQ campaign near Fresno during winter (maroon squares) and from the ArvinO3 study during the summer (green diamonds) versus a proxy of VOC:NO$_x$ ratio estimated by the measured CH$_4$ enhancement over global background divided by the NO$_x$ measured during the flights from the CARB air quality monitoring network nearby.**





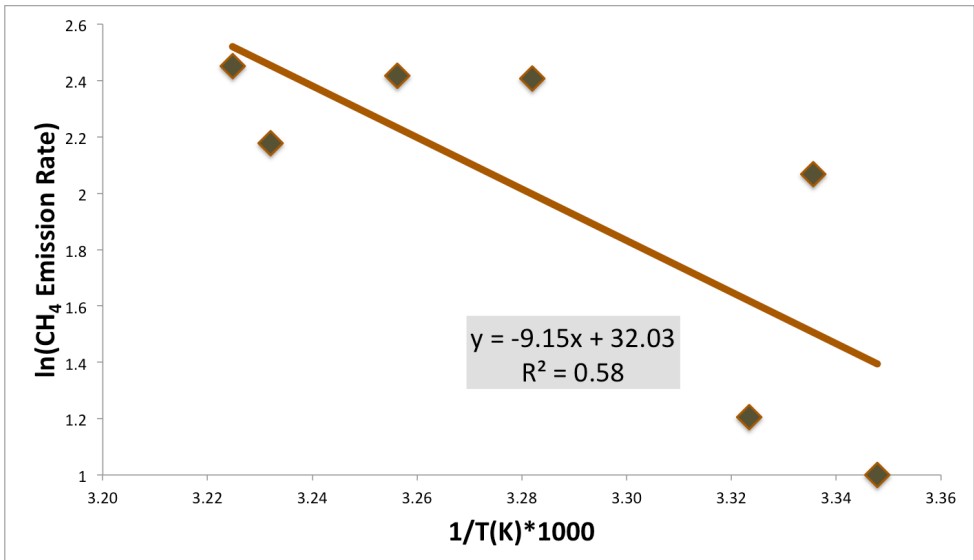

**Figure 10:** Arrhenius plot of the estimated methane emission rate from each flight and the average ABL temperature from the ArvinO3 study where methane emissions are believed to be dominated by agricultural sources.

