# Peer review of "Observing Entrainment Mixing, Photochemical Ozone Production, and Regional Methane Emissions by Aircraft Using a Simple Mixed-Layer Model"

_Atmospheric Chemistry and Physics, 2016_

## Referee Comment (RC1) · Anonymous Referee #1 · 8 Aug 2016

The paper by Trousdell describes new aircraft measurements, which, combined with the mixed layer budget equation, attempts to constrain entrainment, advection, and the emission/production of ozone, methane and water. The dataset and analysis could be suitable for ACP, but as it stands the paper tries to address too many disparate issues: entrainment, the ozone budget, the methane budget, surface heat fluxes and the water cycle. In my opinion the paper needs to be significantly modified before publication.

As a consequence the findings are often not discussed in depth and put into context of uncertainties. A major uncertainty, that needs more evaluation, is the fusion

of in-situ observations with large scale reanalysis data. What are the uncertainties of this approach? E.g. when extracting mean vertical wind speed or surface fluxes from NARR, and plugging these data into eqs. (4),(6), etc., to extract small residuals of the observed quantities. Generally the paper lacks a consistent analysis of error propagation, which makes it hard to follow the uncertainty of the complex method of extracting tracer budgets.

Section 3.2.1: The ozone budget has to be corrected and time-shifted due to rapid photochemistry. Is this done arbitrarily to minimize residuals? Ozone production: methane is used as a VOC tracer to demonstrate that P(O3) is NOx-limited. Yet methane is not a very good tracer, because it has quite different sources compared to VOCs emitted from transport and combustion processes (e.g. aromatics). In addition biogenic VOCs are not considered at all by this approach. Methane is a fugitive emission and therefore does not represent the variation of VOC reactivity properly. To make a more convincing point the authors should use data from the parallel SEACRS mission or ground based observations in combination with a photochemical model to show what fraction of OH reactivity is due to methane (likely very small) and whether methane significantly co-varies with the local VOC reactivity.

In the following section (3.2.2) methane emissions are discussed, but given the uncertainty of the local methane budget (e.g. 100 +/-100Gg/yr), one wonders about the significance of the results. Again, without proper error propagation it makes it hard to follow the validity of the approach, especially uncertainties originating from the model-data fusion. The reader is left with the impression that the approach relies on luck and a fair wind.

Section 3.2.3: Surface latent heat flux: In my opinion this part of the paper presents the most interesting aspects, as it shows a significant bias of surface fluxes obtained from re-analysis data. Why do the authors not present a more in-depth analysis of this finding? Section 4: Rather arbitrarily 5 lines of error analysis are presented here, but only address a very small part that would be necessary for the entire paper.

Generally, in my opinion the paper tries to address too many disparate issues and therefore lacks in depth analysis of the individual pieces. For a focus on ozone, the authors should definitely combine their results with a more comprehensive set of chemistry observations, which seem to be available. For a focus on entrainment and PBL dynamics, a PBL model should be used in conjunction with the budget equation. The paper would also greatly benefit from a more thorough discussion of the associated uncertainties when closing the PBL budget. Perhaps a useful resource to better constrain the thermodynamical and dynamical properties of the PBL during the research flights, and address the propagation of errors and uncertainties, can be found here: http://classmodel.github.io/

---

## Referee Comment (RC2) · Anonymous Referee #2 · 12 Aug 2016

**Review of "Observing Entrainment Mixing, Photochemical Ozone Production, and Regional Methane Emissions by Aircraft Using a Simple Mixed-Layer Model," Trousdell et al., ACP (2016)**

**Summary**

This paper presents results from two small flight campaigns in California. Observed trace gas concentrations and profiles are used to derive entrainment velocities and examine the boundary-layer budgets of ozone, methane and water vapor. Results are used to evaluate photochemical ozone production, regional methane emissions and evapotranspiration.

The presented data is new, and the analysis of boundary layer budgets is a useful technique that is perhaps under-utilized in our field. The paper is generally well-written, although the embellished language is distracting at times and some sections provide an over-abundance of contextual details. Revisions are necessary before publication.

**General Comments**

Section 2.1 provides a wealth of interesting but non-essential details on the topography and meteorology of the SJV. The first three paragraphs could probably be condensed down to one by removing such details –particularly those regarding specific orographic effects, which get confusing unless one constantly refers to a map or is familiar with the area. Indeed, the third paragraph (page 4, line 13) seems totally irrelevant given that the data presented is all daytime. The last paragraph in this section reads like a primer on mountain-valley flows and again seems only tangentially relevant to the results presented later.

The conclusions section is just a summary of main findings. It would be useful to add some discussion of needs for future work, in particular how some of the findings (such as dramatically incorrect emission inventories) could be further verified and eventually incorporated into better emission parameterizations. Is the ABL budget method a practical technique for grounding-truthing regional emissions on a model-relevant scale?

**Specific Comments**

P2/L27: Wolfe et al. (2015) is another relevant and recent citation.

Equations 4-7 and discussion thereof: Seems inconsistent. For example, the surface/entrainment terms are given different symbols for O3 and water. And the entrainment flux sign seems wrong – a higher concentration of stuff in the ABL should give rise to a positive entrainment flux (stuff leaving the ABL) and a negative contribution to dX/dt. It might be more straightforward to show a generic budget equation for any scalar, and then discuss specific treatments for water, ozone and methane.

Page 8, Lines 16-22: suggest deleting.

Eqn. 5: How are the BL concentrations determined for this calculation? Is it an average over the whole ABL, or just the upper portion? Same question for FT? Are uncertainties from this averaging (e.g. std of mean) propagated through to entrainment flux?

P12/L7: how is this map generated? Is it an interpolation of ground site data? Please expound. Also, another way of stating the opposing $O_3$ and $NO_2$ advective terms is that $O_x = O_3 + NO_2$ is conserved.

Section 3.2.3: These findings seem to suggest that NARR has serious flaws and should be adjusted, at least coarsely, to more accurately represent agricultural practices in some broad sense. A naïve question: would such issues impact the subsidence velocity derived from NARR?

Table 3: The third column is technically not a flux, but a flux divergence. Also, please give $CH_4$ production in ppmv/h for easy comparison with other terms.

Figure 9: is there any physical rationale behind a power-law fit?

**Technical Comments**

Fig. 2: Please label flight regions 1 and 2 as referenced in section 2.1.

RASS is defined twice.

P6/L32: delete ", which"

P6/L35: "as per the Fundamental Theorem of Calculus" is a gratuitously pretentious statement.

Equations 1-3: subsidence is referred to as both $W(z_i)$ and $W$. Pick one.

P9, L13: delete "the 5 hour period  of late morning to early afternoon from"

P10/L17: delete "a remove of"

---

## Referee Comment (RC3) · Anonymous Referee #3 · 3 Sep 2016

Review: Observing Entrainment Mixing, Photochemical Ozone Production, and Regional Methane Emissions by Aircraft Using a Simple Mixed-Layer Model

This paper describes the design and execution of two flight experiments in the San Joaquin Valley of California to quantify entrainment rates and then uses these entrainment velocities to solve for: (a) ozone production rates, (b) methane emissions, and (c) evapotranspiration. The authors are attempting numerous things here, which makes the paper difficult to read and, at times, the results difficult understand. The work is interesting, but paper would benefit from better organization around a clear goal prior

to publication. Adding clarity may be as simple as removing the excessive inessential detail.

General comments:

The Introduction should be reorganized to better frame the work. Some specific issues are as follows. In paragraph 2, the text does not define "tracer method" or "budget of the inversion base height" when describing what is done in the forthcoming analysis. This makes it difficult for the reader to know what is done here and how this work is different from past work. The sentence, "by way of targeted airborne campaigns we are able to probe the regional ABL vertically and horizontally and calculate entrainment rates and mesoscale advection," seems key, but is placed awkwardly in the middle of paragraph 3. The fourth paragraph returns to the idea of scalar budgeting, but still does not define, instead suggesting I should already be familiar with the concept (done through the particular way the references are discussed). While I agree with the content in paragraph 5, this paper is not actually about, "better understand[ing] the diurnal behavior of the wintertime boundary layer in the San Joaquin Valley." The discussion in paragraph 6 should more relevant to the analysis performed. For example, the paper never significantly discusses PM, but investigates ozone production, methane emissions, and evapotranspiration. While there is some text on ozone and drought here, methane is absent entirely. The last paragraph presents an outline of the paper, but the preceding text has not setup these goals, nor does the outline mention the ozone production, methane emission, or evapotranspiration applications.

Most of Section 2.1 is irrelevant. The authors should relate the descriptive information directly back to their analysis and delete superfluous detail.

Sections 2.6 and 2.7 should be framed around what was done here, rather than as done currently, as a general discussion of the two methods using the author's dataset as an example. The last sentence of Section 2.7, "ultimately the approach using the budget of boundary layer inversion height, outlined in Section 2.6 was taken to calculate

the entrainment rate," should be given to the reader up front. Additionally, the last paragraph in 2.7 is described almost narratively of how the analysis was done. Please reorder such that results are presented to convey the logic of the analysis to the reader.

What are the results for Ox, as opposed to O3 and NO2 separately? Use of P(Ox) would be especially important in the wintertime and better suited for a winter/summer comparison. Secondly, has wintertime P(O3) been found to be NOx-limited also? That seems unlikely; please clarify.

Broadly, the outline of the paper is to compute the entrainment rate and then use this rate to explore three things: (a) ozone production rates, (b) methane emissions, and (c) water. Adding text or a dedicated section after discussion of the three studies, but prior to the Conclusion, that ties everything back together would do two valuable things. First, it would clarify the narrative and logic of the paper, and second, it would reinforce the significance of the work.

Specific comments:

Page 2, lines 3–4: Citation needed on, "this mixing tends to be a significant contributor to the ABL budget of the scalar."

Page 3, lines 17–18: Should this be 105 exceedances "per year"?

Page 7, line 7: w(e) is not defined in the text (it is instead defined on page 8, line 23).

Page 10, lines 18–20: What is the evidence for: "For the purposes of estimating regional source strengths or regional in situ photochemistry, we suggest that the more pertinent mixing process is the dilution of the anthropogenically influenced ABL air mass by the more global 'baseline' FT air."

Page 11, lines 34–35: How is this shown in Fig. 7: "the importance of entrainment mixing on an ozone exceedance day."

Page 12, lines 35–36: It is difficult to see that methane is an appropriate proxy for

total VOC. Even if dairies and gas production are the dominant source of VOCs, what matters more is that the drivers of methane emission match the drivers of the other VOC, which might not be true even if the sources are the same.

Page 13, lines 3–5: Can an estimate of the uncertainty be given?

Section 4: I recommend moving Section 4 to precede Sections 3.2.1–3.2.3.

---

## Author Comment (AC1) · 22 Sep 2016

(Referee)The paper by Trousdell describes new aircraft measurements, which, combined with the mixed layer budget equation, attempts to constrain entrainment, advection, and the emission/production of ozone, methane and water. The dataset and analysis could be suitable for ACP, but as it stands the paper tries to address too many disparate issues: entrainment, the ozone budget, the methane budget, surface heat fluxes and the water cycle. In my opinion the paper needs to be significantly modified before publication. As a consequence the findings are often not discussed in depth

and put into context of uncertainties.

(Response)Thank you for spending the time to go through our paper, your comments and critiques are greatly appreciated. I understand how you could see that we attempt to take on disparate issues, but our goal is precisely that: to bridge dynamics and chemistry. We feel that the atmospheric chemistry and boundary layer communities can benefit from each other, and the use of this simple mixed layer model demonstrates that. This journal attracts those interested in atmospheric dynamics as well as chemistry so we believe it is the perfect fit for our manuscript. To clarify, our intention when detailing these various topics like the ozone budget, the methane budget, entrainment, etc. is not to necessarily go into great depths on each but to show how a simple mixed layer budget equation sufficiently closed by in-situ flight data, including a detailed calculation of entrainment, can be used to uncover useful and novel estimates of emissions and photochemical rates. With this in mind, and in light of the complex mesoscale environment, we feel it is inadvisable to to add all of the details from these various topics, yet it is important to present them together. The crux of this study is really the computation of dynamic quantities, like entrainment and linking them to the chemistry of the boundary layer.

On the other hand, we can see how our treatment of the uncertainties of these estimates could come across as lacking depth, so we have rewritten that entire section (4 Error Analysis) to clarify our estimates of the uncertainty of this approach. A major uncertainty, that needs more evaluation, is the fusion of in-situ observations with large scale reanalysis data. What are the uncertainties of this approach? E.g. when extracting mean vertical wind speed or surface fluxes from NARR, and plugging these data into eqs. (4),(6), etc., to extract small residuals of the observed quantities. To be clear, we do not put surface fluxes nor mean vertical velocities into equations 4 and 6. The only reanalysis data we incorporate is into equation 2, the inversion height budget equation. To answer the question directly, I would refer the reviewer to Table 1, wherein it appears that the very conservative uncertainty we assign to the mean vertical ve-

locity of the NARR (0.5 cm/s or approximately 50%) leads to large uncertainties in the derived entrainment velocities: ~1.0 cm/s for each project average (1.5 and 3.0 cm/s averages). We feel this is a reasonable estimate of this uncertainty and do propagate it through the entrainment terms in the other budget equations (4 & 6) where that term is not always the leading one, however the large uncertainties in methane emissions are a direct result of these assumed uncertainties. Therefore, we disagree with the reviewer in that this is not a case of trying to tease out a small residual from large terms with large uncertainties.

With regards to the suitability of our estimated uncertainty in the NARR vertical velocities, we refer to a study by Albrecht et al. (2016). They also utilized reanalysis data in the form of omega, which was later transformed to vertical velocity and subsequently used in the exact same inversion height budget equation we use. They estimated the error of the average vertical velocity derived in this fashion to be $\pm$ 0.1 cm s-1, a full factor of five times smaller than ours. In addition, they conclude that the majority of variation from this budget equation is reflected in the local time rate of change of inversion height when compared to the variations in the advection of inversion height and vertical velocity combined. We also found the budget equation of inversion height to be dominated by time rate of change on average, so feel that we are measuring the most important term in the $z_i$ governing equation (2). We have added these details to our error analysis section to help clarify these points.

(Referee)Generally the paper lacks a consistent analysis of error propagation, which makes it hard to follow the uncertainty of the complex method of extracting tracer budgets. We have expanded our error analysis section (Section 4) to include a more detailed analysis. Some of the errors were calculated formally using a standard error which is a residual from the linear fit normalized by the number of data points, but other error terms are not subject to such statistical formalism. For instance, the error in the scalar jump, which is diagnosed by eye from vertical profiles is given what we deem a conservative estimate of its error. We also note that our estimated error for vertical

velocity obtained from NARR is five times greater than that used by the Albrecht et al 2016 study for the reanalysis data they used (ECMWF). We try to be careful and we include errors with every term in the budgets. For cases when terms were not subject to a rigorous mathematical analysis of error we attempt to be conservative and overestimate the potential error. In Tables 1-3 we have now included the standard deviations of the mean values from each campaign to give a sense of the natural, background variation relative to the observational uncertainty estimates to help place these in context. Section 3.2.1: The ozone budget has to be corrected and time-shifted due to rapid photochemistry. Is this done arbitrarily to minimize residuals?

(Response)Our ozone budgets were not time shifted or corrected for rapid photochemistry. The only instance in the manuscript where we corrected the ozone levels was to make plots of horizontal gradients and advection, which were corrected by way of the secular linear time rate of change to a common time stamp. This reduces the spatial 'noise' of the aircraft measurements which are sweeping over the region throughout the day when the mean ozone is on the rise. Ozone production: methane is used as a VOC tracer to demonstrate that P(O3) is NOx-limited. Yet methane is not a very good tracer, because it has quite different sources compared to VOCs emitted from transport and combustion processes (e.g. aromatics). In addition biogenic VOCs are not considered at all by this approach. Methane is a fugitive emission and therefore does not represent the variation of VOC reactivity properly. To make a more convincing point the authors should use data from the parallel SEACRS mission or ground based observations in combination with a photochemical model to show what fraction of OH reactivity is due to methane (likely very small) and whether methane significantly co-varies with the local VOC reactivity. As discussed in Section 3.2.2 the majority of methane in both studies are believed to be associated with fossil fuel extraction and dairy operations. The studies of Gentner et al. [2014] and Pusede et al. [2014] indicate that methane is fairly well correlated with alcohols (which have strong dairy sources), higher alkanes (natural gas), and CO (other anthropogenic activities.) While we acknowledge that methane is a somewhat crude tracer of reactive VOC, we present the results because

there is a suggestive relationship with our inferred ozone production rates that is consistent with past studies of the ozone production regime.

With respect to the SEAC4RS dataset we found only one boundary layer leg within the Central Valley of California during that mission. With that we have about one hour of data taken in the early evening containing 28 data points from the dataset of Don Blake showing a correlation of 0.6 or greater with CH4 for; CO, DMS, HCFC-124, HFC-134a, HFC-152a, CH3I, CH2Cl2, C2HCl3, C2Cl4, MeONO2, EtONO2, i-PrONO2, n-PrONO2, 2-BuONO2, 3-Methyl-2-BuONO2, 3-PenONO2, 2-PeONO2, Ethane, Ethene, Ethyne, Propane, Propene, n-Butane, 1-Butene, i-Butene, i-Pentane, n-Pentane, 1-Pentene, 2_3-Dimethylbutane, 2-Methylpentane, 3-Methylpentane, n-Heptane, Benzene, Ethylbenzene, and beta-Pinene. From this mix of hydrocarbons we maintain that CH4 is a decent, although imperfect, tracer for other reactive hydrocarbons. Trying to use this limited data set in a photochemical model seems well beyond the scope of this paper, and the NOx-limited nature of the ozone environment has been confirmed in other studies (Pusede et al. [2012], Brune et al., [2016]).

(Referee)In the following section (3.2.2) methane emissions are discussed, but given the un- certainty of the local methane budget (e.g. 100 +/-100Gg/yr), one wonders about the significance of the results. Again, without proper error propagation it makes it hard to follow the validity of the approach, especially uncertainties originating from the model- data fusion. The reader is left with the impression that the approach relies on luck and a fair wind.

(Response)It is true our methane emission errors are of a similar order of magnitude as the overall flight-to-flight spread. This is caused by the fact that we estimate our errors in entrainment to be nearly the same order of magnitude as the results. But this magnitude of uncertainty in entrainment is common for measurements of such a difficult, yet important, parameter (de Arellano et al. [2004] de Roode & Duynkerke, [1997]; Bretherton et al, [1995]; Wolfe et al., 2015). So, naturally emission estimates that are derived directly from this parameter are going to have similarly large uncertainties. But

we believe that it is still a valid measurement, and when repeated over many flights, the mean measurement is indeed meaningful. Furthermore, this is a very important result to the methane community, which is faced with a paucity of such estimates. It also might be useful to note here that inverse modeling techniques used to derive a posteriori emission estimates likely have similarly large uncertainties, but these are rarely, if ever, explicitly treated in modeling papers (Cui et al. 2015).

(Referee)Section 3.2.3: Surface latent heat flux: In my opinion this part of the paper presents the most interesting aspects, as it shows a significant bias of surface fluxes obtained from re-analysis data. Why do the authors not present a more in-depth analysis of this finding?

(Response)The calculation of the water budget was an easy addition for us because our payload measures water vapor, and since the budget equation for water does not have any internal source terms under our flight conditions. The results are included to show the robustness and wide applicability of the budget method. We agree the findings are interesting, but we leave them as general warnings to the community about the latent heat calculated in NARR, and that this is certainly going to have an effect on ABL heights due to partitioning of latent vs. sensible heat fluxes. But to probe this result more deeply would require a lot more information about the land-surface and we feel would distract from the main objective of the manuscript.

(Referee)Section 4: Rather arbitrarily 5 lines of error analysis are presented here, but only address a very small part that would be necessary for the entire paper.

(Response)We agree with you that our error analysis was overly concise, and we hope that the expanded error analysis section will assuage many of your concerns. Generally, in my opinion the paper tries to address too many disparate issues and therefore lacks in depth analysis of the individual pieces. For a focus on ozone, the authors should definitely combine their results with a more comprehensive set of chemistry observations, which seem to be available. We believe that estimating net O3 production

is a significant feat, and we have done so with equal or better uncertainty than other reports of in the literature (Pusede et al, 2014; Brune et al., 2016). Furthermore, without a vast array of chemical species and meteorological data to constrain a model, we do not feel that all that much would be gained in such an exercise.

(Referee)For a focus on entrainment and PBL dynamics, a PBL model should be used in conjunction with the budget equation. The paper would also greatly benefit from a more thorough discussion of the associated uncertainties when closing the PBL budget. Perhaps a useful resource to better constrain the thermodynamical and dynamical properties of the PBL during the research flights, and address the propagation of errors and uncertainties, can be found here: http://classmodel.github.io/

(Response)We hope that in light of our responses here and above the reviewer will reconsider their conclusion that the absence of applying more complicated models to this data set is a sign of a superficial treatment of the subject. Our intention here is to present an empirical study of surface emissions, ozone photochemistry, and entrainment in the San Joaquin Valley, and the wide applicability of the airborne budget method we have applied. Perhaps our use of the term 'model' in the title is a bit misleading, because by 'model' we really mean a simple analytical tool that can be applied to airborne data. We do not wish to resort to any higher order models in this analysis, because such models necessarily require boundary conditions and initial conditions that were not constrained by observation – specifically, OH reactivity and/or speciated VOC data in the ozone analysis, and surface heat fluxes in the entrainment analysis. We do not believe that the particular model referred to above will help us better understand uncertainty, but will rather add more. The Dutch slab model is based on many inputs including, but not limited to, surface heat fluxes, drag coefficients, initial boundary layer height, free tropospheric stability, and of course subsidence.. Additionally, the model does not include advection so we would still need some way to account for the uncertainties of this term and the subsidence term, but would have no way to know the uncertainties in the surface heat or momentum fluxes. We feel the method presented

here is more direct because it is not driven by all of these unknown parameters and more closely tracks the uncertainties in the governing equations themselves, as we hope is now more clearly presented in the new section 4.

References:

Albrecht, B., Fang, M., Ghate, V.: Exploring Stratocumulus Cloud-Top Entrainment Processes and Parameterizations by Using Doppler Cloud Radar Observations, J. Atmos. Sci. 73(2), 729-42, 2016.

Bretherton, C. S., and Pincus, R.: CLOUDINESS AND MARINE BOUNDARY-LAYER DYNAMICS IN THE ASTEX LAGRANGIAN EXPERIMENTS .1. SYNOPTIC SETTING AND VERTICAL STRUCTURE, J. Atmos. Sci., 52, 2707-2723, 10.1175/1520-0469(1995)052<2707:cambld>2.0.co;2, 1995.

Cui, Y. Y., Brioude, J., McKeen, S. A., Angevine, W. M., Kim, S. W., Frost, G. J., Ahmadov, R., Peischl, J., Bousserez, N., Liu, Z., Ryerson, T. B., Wofsy, S. C., Santoni, G. W., Kort, E. A., Fischer, M. L., and Trainer, M.: Top-down estimate of methane emissions in California using a mesoscale inverse modeling technique: The South Coast Air Basin, J. Geophys. Res.-Atmos., 120, 6698-6711, 10.1002/2014jd023002, 2015.

deArellano, Vilà-Guerau, J., Gioli, B., Miglietta, F., Jonker, H. J. J., Baltink, H. K., Hutjes, R. W. A. and Holtslag, A. A. M.: Entrainment Process of Carbon Dioxide in the Atmospheric Boundary Layer, J. Geophys. Res. 109(D18), 2004. Elsgaard, L., Olsen, A. B., and Petersen, S. O.: Temperature response of methane production in liquid manures and co-digestates, Sci Total Environ, 539, 78-84, 10.1016/j.scitotenv.2015.07.145, 2016.

deRoode, S. R., and Duynkerke, P. G.: Observed Lagrangian transition of stratocumulus into cumulus during ASTEX: Mean state and turbulence structure, J. Atmos. Sci., 54, 2157-2173, 10.1175/1520-0469(1997)054<2157:oltosi>2.0.co;2, 1997.

Gentner, D. R., Ford, T. B., Guha, A., Boulanger, K., Brioude, J., Angevine, W. M., de

[Figure]

Gouw, J. A., Warneke, C., Gilman, J. B., Ryerson, T. B., Peischl, J., Meinardi, S., Blake, D. R., Atlas, E., Lonneman, W. A., Kleindienst, T. E., Beaver, M. R., St Clair, J. M., Wennberg, P. O., VandenBoer, T. C., Markovic, M. Z., Murphy, J. G., Harley, R. A., and Goldstein, A. H.: Emissions of organic carbon and methane from petroleum and dairy operations in California's San Joaquin Valley, Atmospheric Chemistry and Physics, 14, 4955-4978, 10.5194/acp-14-4955-2014, 2014.
* * *

---

## Author Comment (AC2) · 22 Sep 2016

(Response)Thank you for your time and critiques of our work; they are greatly appreciated. We feel it will be clearest to respond to each bullet point made by including it as a blue comment beneath each respective point.

(Referee)Review of "Observing Entrainment Mixing, Photochemical Ozone Production, and Regional Methane Emissions by Aircraft Using a Simple Mixed-Layer Model," Trousdell et al., ACP (2016) Summary This paper presents results from two small flight campaigns in California. Observed trace gas concentrations and profiles are used

to derive entrainment velocities and examine the boundary-layer budgets of ozone, methane and water vapor. Results are used to evaluate photochemical ozone production, regional methane emissions and evapotranspiration. The presented data is new, and the analysis of boundary layer budgets is a useful technique that is perhaps under-utilized in our field. The paper is generally well-written, although the embellished language is distracting at times and some sections provide an over-abundance of contextual details. Revisions are necessary before publication. General Comments Section 2.1 provides a wealth of interesting but non-essential details on the topography and meteorology of the SJV. The first three paragraphs could probably be condensed down to one by removing such details –particularly those regarding specific orographic effects, which get confusing unless one constantly refers to a map or is familiar with the area. Indeed, the third paragraph (page 4, line 13) seems totally irrelevant given that the data presented is all daytime. The last paragraph in this section reads like a primer on mountain-valley flows and again seems only tangentially relevant to the results presented later.

(Response)We understand the referee's point here, and we have condensed much of the information originally presented. We chose to include a clear survey of mountain-valley dynamics to set the stage for this unique mesoscale environment in which we are working and because we do not find such a concise treatment in the extant literature. It is exactly this dynamically complex environment which has exacerbated the markedly poor air quality in the region. For others working on the recalcitrant air quality issues in this area, or similar ones such as the Po Valley in Italy, we feel this information is essential for consideration.

(Referee)The conclusions section is just a summary of main findings. It would be useful to add some discussion of needs for future work, in particular how some of the findings (such as dramatically incorrect emission inventories) could be further verified and eventually incorporated into better emission parameterizations. Is the ABL budget method a practical technique for grounding-truthing regional emissions on a modelrelevant scale?

(Response)We have add two paragraphs to the conclusions in order to suggest further research that may build on the accomplishments of this study.

(Referee)Specific Comments P2/L27: Wolfe et al. (2015) is another relevant and recent citation.

(Response)Thank you, yes, we have added that reference at this point. We had already included it in our paper elsewhere but had neglected it here.

(Referee)Equations 4-7 and discussion thereof: Seems inconsistent. For example, the surface/entrainment terms are given different symbols for O3 and water. And the entrainment flux sign seems wrong – a higher concentration of stuff in the ABL should give rise to a positive entrainment flux (stuff leaving the ABL) and a negative contribution to dX/dt. It might be more straightforward to show a generic budget equation for any scalar, and then discuss specific treatments for water, ozone and methane.

(Reponse)You are correct, equation 4 had a sign inconsistency from our other equations, and there was substantial inconsistency in the symbols we had used. We have more systematically applied consistent symbols for the scalar budget equations and corrected the sign mistake. In response to a perceived misconception apparent in the reviewer's comment, we further added some discussion to clarify the role of entrainment in the ABL budget equations. A higher ABL concentration with everything else fixed would give rise to a dilution of the boundary layer concentrations, and yes this drives a negative dC/dt. However, this is not due to "stuff leaving the ABL" as the reviewer states. Entrainment in an actively turbulent ABL is an irreversible mixing process that incorporates free tropospheric (FT) air into the ABL, not vice versa. The positive scalar flux at the ABL top is the equivalent to a downward flux of concentration deficit (when the FT possesses a lower concentration), and we have explicitly stated that in the text now. We thank the reviewer for bringing this to our attention.

(Referee)Page 8, Lines 16-22: suggest deleting.

(Response)Advice taken; we have removed these lines from the manuscript. We originally wanted to emphasize that in principal different scalars could be used in their respective budget equation to expose entrainment rates, i.e. water, ozone, or methane, and have made that point up front during the discussion of equations 4-7 as per reviewer's suggestions.

(Referee)Eqn. 5: How are the BL concentrations determined for this calculation? Is it an average over the whole ABL, or just the upper portion? Same question for FT? Are uncertainties from this averaging (e.g. std of mean) propagated through to entrainment flux?

(Response)The scalar jump is determined from looking at vertical profiles and making the best eye judgment of the difference in concentrations between the top half of the ABL and the lowest ~100m of the FT. Often it is quite clear as can be seen in our example from fig. 7. We have included a brief description of how these values are determined and their estimated uncertainties, which are like all the terms propagated through to the final results. The error analysis section (4) has been greatly expanded so this should be much clearer now.

(Referee)P12/L7: how is this map generated? Is it an interpolation of ground site data? Please expound. Also, another way of stating the opposing O3 and NO2 advective terms is that Ox=O3+NO2 is conserved.

(Response)The NOx and O3 advective maps are interpolated to a 2D grid from aircraft data taken in the ABL. All data is corrected for the calculated mean regional time rate of change back to a common time stamp of 13:30. This has been more clearly explicated in the text. As for the odd oxygen interpretation, we do not agree. The gradient of ozone is an order of magnitude greater than that of NO2. This is not simply a titration situation, but is intimately linked to rapid ozone production. We feel that the discussion of odd oxygen in this study would not serve to illuminate because it introduces a further

unknown variable of the NOx emission rates. Also, we only had the NO2 measurement on one single flight.

(Referee)Section 3.2.3: These findings seem to suggest that NARR has serious flaws and should be adjusted, at least coarsely, to more accurately represent agricultural practices in some broad sense. A naïve question: would such issues impact the subsidence velocity derived from NARR?

(Response)We do not believe that large scale vertical motion would be all that susceptible to partitioning of surface heat fluxes among latent to sensible, but it certainly affects the convective activity and entrainment and boundary layer depths in the model. Subsidence is generally believed to be controlled by synoptic flow conditions. Although we do suspect that subsidence can modified a good bit due to mesoscale orography. A better representation of agricultural practices would lead to a better estimate of the latent heat flux, which affects the partitioning in the surface buoyancy flux, and for a constant net radiation forcing this would lead to lower ABL heights for greater latent heat fluxes. This is why the NARR ABL depths are so much higher than measured, for instance.

(Referee)Table 3: The third column is technically not a flux, but a flux divergence. Also, please give CH4 production in ppmv/h for easy comparison with other terms.

(Response)The third column is the entrainment flux contribution to the flux divergence. We report it that way to have it in comparable units to the other terms. But reporting the surface emission similarly would not make sense to us, as the units most people are familiar with are something like the chosen ones of Gigagrams per year. The CH4 production (surface emission) term is simply the numerical sum of the other columns, so we thought it would be redundant to see it in the same units.

(Referee)Figure 9: is there any physical rationale behind a power-law fit?

(Response)The short answer is no. We know that the ozone chemistry is non-linear,

and the simplest non-linear relationship is a power law.

(Referee)Technical Comments Fig. 2: Please label flight regions 1 and 2 as referenced in section 2.1.

(Response)We have changed the legend of Figure 2 to indicate the region numbers 1 and 2.

(Referee)RASS is defined twice. (Response)Got it.

(Referee)P6/L32: delete ", which" (Response)Deleted

(Referee)P6/L35: "as per the Fundamental Theorem of Calculus" is a gratuitously pretentious statement.

(Response)We did not consider that such a foundational mathematical principle could be considered pretentious, but have eliminated the wording to protect the common reader.

(Referee)Equations 1-3: subsidence is referred to as both $W(z_i)$ and $W$. Pick one.

(Response)Okay, thanks we will stick with just, $W$, with the implicit understanding that it can be a strong function of height.

(Referee)P9, L13: delete "the 5 hour period of late morning to early afternoon from" P10/L17: delete "a remove of"

(Response)Done.

---

## Author Comment (AC3) · 22 Sep 2016

(Referee)Review: Observing Entrainment Mixing, Photochemical Ozone Production, and Regional Methane Emissions by Aircraft Using a Simple Mixed-Layer Model

This paper describes the design and execution of two flight experiments in the San Joaquin Valley of California to quantify entrainment rates and then uses these entrainment velocities to solve for: (a) ozone production rates, (b) methane emissions, and (c) evapotranspiration. The authors are attempting numerous things here, which makes the paper difficult to read and, at times, the results difficult understand. The work is

interesting, but paper would benefit from better organization around a clear goal prior to publication. Adding clarity may be as simple as removing the excessive inessential detail.

General comments:

The Introduction should be reorganized to better frame the work. Some specific issues are as follows. In paragraph 2, the text does not define "tracer method" or "budget of the inversion base height" when describing what is done in the forthcoming analysis. This makes it difficult for the reader to know what is done here and how this work is different from past work.

(Response)We have added some clarification clauses to describe these methodologies, but exact details have to be postponed to the method descriptions of Section 2.

(Referee)The sentence, "by way of targeted airborne campaigns we are able to probe the regional ABL vertically and horizontally and calculate entrainment rates and mesoscale advection," seems key, but is placed awkwardly in the middle of paragraph 3.

(Response)This statement is made after introducing the concepts of entrainment and advection, and therefore does not seem awkward in its placement to us. We have attempted to make a more clarion statement of the paper's overarching goal at the end of paragraph 3, keeping in mind that positional emphasis is typically carried by the end sentence of a paragraph (The Elements of Style, by Strunk & White [1999]):

The central goal of the work presented here is to show how, by way of targeted small-scale airborne campaigns, it is possible to probe the regional ABL vertically and horizontally to calculate entrainment rates and mesoscale advection, and thereby shed light on all of the processes that change the concentrations of trace gases in the boundary layer throughout the day. This methodology thereby reveals the quantitative origins

of chemical constituents measured in near-surface air, by comparing direct observations of all but one of the leading terms of the scalar budget equation, and inferring the unknown term as a residual.

(Referee)The fourth paragraph returns to the idea of scalar budgeting, but still does not define, instead suggesting I should already be familiar with the concept (done through the particular way the references are discussed).

(Response)We have defined a scalar budget in an added subordinate clause in the second paragraph, and a new sentence at the end of the third paragraph as per earlier suggestions. Then we devote the entirety of Section 2.7 to defining exactly what the methodology is. We do not see how to further clarify the technique in the introduction without burdening the section with excessive detail.

(Referee)While I agree with the content in paragraph 5, this paper is not actually about, "better understand[ing] the diurnal behavior of the wintertime boundary layer in the San Joaquin Valley."

(Response)We think that reporting observed entrainment rates in the winter, which have never been reported, does in fact help to better understand the ABL's diurnal behavior.

(Referee)The discussion in paragraph 6 should more relevant to the analysis performed. For example, the paper never significantly discusses PM, but investigates ozone production, methane emissions, and evapotranspiration. While there is some text on ozone and drought here, methane is absent entirely.

(Response)We have added a concluding sentence to this paragraph that helps to establish the importance of the work: Entrainment aloft becomes an even more important factor during stagnant conditions in the SJV because it represents the principal mode of ventilating the air pollutants in the ABL, and therefore its quantification is crucial to predicting the intensity and duration of an air quality episode. Although the work does

not explicitly address PM issues, the results are directly applicable to the wintertime PM problem in the SJV and we hope will be used by others working on the DISCOVER-AQ data set. Also, because methane is not directly an air quality concern, we leave it out of this paragraph. We have removed a couple of sentences in the hopes that they might be considered "excessive inessential details."

(Referee)The last paragraph presents an outline of the paper, but the preceding text has not setup these goals, nor does the outline mention the ozone production, methane emission, or evapotranspiration applications.

(Response)We have expanded the outline paragraph in an attempt to state the goals of our work more clearly, as per the reviewer's earlier suggestion.

(Referee)Most of Section 2.1 is irrelevant. The authors should relate the descriptive information directly back to their analysis and delete superfluous detail. (Response)We have condensed much of the information originally presented in Section 2.1 as it was also suggested by reviewer 2. However, we disagree that this discussion of the dynamic environment is irrelevant. We chose to include a clear survey of mountain-valley dynamics to set the stage for this unique mesoscale environment in which the experiments took place and because we do not find such a concise description anywhere in the extant literature. This dynamic complexity lies at the heart of why the region endures some of the poorest air quality in the nation. For others working on recalcitrant air quality issues in this area, or similar ones such as the Po Valley in Italy, we feel this information is essential for consideration.

(Referee)Sections 2.6 and 2.7 should be framed around what was done here, rather than as done currently, as a general discussion of the two methods using the author's dataset as an example. The last sentence of Section 2.7, "ultimately the approach using the budget of boundary layer inversion height, outlined in Section 2.6 was taken to calculate the entrainment rate," should be given to the reader up front. Additionally, the last paragraph in 2.7 is described almost narratively of how the analysis was done.

Please reorder such that results are presented to convey the logic of the analysis to the reader.

(Response)We have restructured/rewritten Section 2.7 to better coordinate the general discussion of the scalar budget equations with how they were used in these experiments.

(Referee)What are the results for Ox, as opposed to O3 and NO2 separately? Use of P(Ox) would be especially important in the wintertime and better suited for a winter/summer comparison. Secondly, has wintertime P(O3) been found to be NOx-limited also? That seems unlikely; please clarify.

(Response)Unfortunately, we did not have measurements of NO2 save for one single flight, and therefore were not able to perform a budget of odd oxygen.

- Yes, the results presented in Fig. 9 indicate that P(O3) is NOx-limited in the wintertime, but the inference is not strong given the limited spread in VOC:NOx ratio, and the uncertainties in using CH4 as a general VOC proxy. Nevertheless, we feel the result is worth presenting, especially since very little is known about winter O3 production because it is not often considered.

(Referee)Broadly, the outline of the paper is to compute the entrainment rate and then use this rate to explore three things: (a) ozone production rates, (b) methane emissions, and (c) water. Adding text or a dedicated section after discussion of the three studies, but prior to the Conclusion, that ties everything back together would do two valuable things. First, it would clarify the narrative and logic of the paper, and second, it would reinforce the significance of the work.

(Response)We have attempted to tie everything together more clearly throughout the revised manuscript and thus do not see the value in repeating this before doing so again in the conclusions.

(Referee)Specific comments:
Page 2, lines 3–4: Citation needed on, "this mixing tends to be a significant contributor to the ABL budget of the scalar."

(Response)Stull [1990], Arellano et al. [2011], Lehning et al. [1998].

(Referee)Page 3, lines 17–18: Should this be 105 exceedances "per year"?

(Response)We have eliminated this statement as non-essential.

(Referee)Page 7, line 7: w(e) is not defined in the text (it is instead defined on page 8, line 23).

(Response)Defined in both places now.

(Referee)Page 10, lines 18–20: What is the evidence for: "For the purposes of estimating regional source strengths or regional in situ photochemistry, we suggest that the more pertinent mixing process is the dilution of the anthropogenically influenced ABL air mass by the more global 'baseline' FT air."

(Response)This is more of a conjecture, claiming that it is the ABL growth rate after its initial 'encroachment' through the morning's residual layer that is key in understanding regional chemistry and surface emissions because the residual layer tends to be made up of mostly recycled air from the region. Of course, in principle, the budgets should still hold during the more rapid growth of the morning ABL, but they become more difficult to accurately measure due to the greater presence of transients and inhomogeneities. We do not feel this detail should be introduced into the manuscript because it is somewhat tangential as we did not perform the budget analysis in the morning hours, and it would not make sense to anyway because of the low O3 production at high solar zenith angles, which does not impact the afternoon O3 maximum very significantly.

(Referee)Page 11, lines 34–35: How is this shown in Fig. 7: "the importance of entrainment mixing on an ozone exceedance day."

(Response)It is shown in the subsequent discussion where the jumps observed in Fig.

7 are used to estimate a time rate of change of O3 and NO2 concentrations due to entrainment dilution.

(Referee)Page 12, lines 35–36: It is difficult to see that methane is an appropriate proxy for total VOC. Even if dairies and gas production are the dominant source of VOCs, what matters more is that the drivers of methane emission match the drivers of the other VOC, which might not be true even if the sources are the same.

(Response)As discussed in Section 3.2.2 the majority of methane in both studies are believed to be associated with fossil fuel extraction and dairy operations. The studies of Gentner et al. [2014] and Pusede et al. [2014] indicate that methane is fairly well correlated with alcohols (which have strong dairy sources), higher alkanes (natural gas), and CO (other anthropogenic activities.) While we acknowledge that methane is a somewhat crude tracer of reactive VOC, we present the results because there is a suggestive relationship with our inferred ozone production rates that is consistent with past studies of the ozone production regime.

(Referee)Page 13, lines 3–5: Can an estimate of the uncertainty be given?

(Response)We have included an average uncertainty estimate from our experimental results to better frame the comparison, and have done so in all of the Tables as well. There is no estimate of uncertainty in P(O3) made by Pusede et al. (2014).

(Referee)Section 4: I recommend moving Section 4 to precede Sections 3.2.1–3.2.3.

(Response)We feel that a discussion of the errors in the measurements specifics is best delayed until the details of the experimental results are related, so we have kept Section 4 after Section 3, but we have expanded it considerably to make clear exactly how our errors have been treated in our results.

---

## Author Comment (AC5) · 27 Sep 2016

Note* all responses are in blue type.

Thank you for your time and critiques of our work; they are greatly appreciated. We feel it will be clearest to respond to each bullet point made by including it as a blue comment beneath each respective point.

Review of "Observing Entrainment Mixing, Photochemical Ozone Production, and Regional Methane Emissions by Aircraft Using a Simple Mixed-Layer Model," Trousdell et al., ACP (2016)

Summary

This paper presents results from two small flight campaigns in California. Observed trace gas concentrations and profiles are used to derive entrainment velocities and examine the boundary-layer budgets of ozone, methane and water vapor. Results are used to evaluate photochemical ozone production, regional methane emissions and evapotranspiration.

The presented data is new, and the analysis of boundary layer budgets is a useful technique that is perhaps under-utilized in our field. The paper is generally well-written, although the embellished language is distracting at times and some sections provide an over-abundance of contextual details. Revisions are necessary before publication.

General Comments

Section 2.1 provides a wealth of interesting but non-essential details on the topography and meteorology of the SJV. The first three paragraphs could probably be condensed down to one by removing such details –particularly those regarding specific orographic effects, which get confusing unless one constantly refers to a map or is familiar with the area. Indeed, the third paragraph (page 4, line 13) seems totally irrelevant given that the data presented is all daytime. The last paragraph in this section reads like a primer on mountain-valley flows and again seems only tangentially relevant to the results presented later.

We understand the referee's point here, and we have condensed much of the information originally presented.  We chose to include a clear survey of mountain-valley dynamics to set the stage for this unique mesoscale environment in which we are working and because we do not find such a concise treatment in the extant literature. It is exactly this dynamically complex environment which has exacerbated the markedly poor air quality in the region. For others working on the

recalcitrant air quality issues in this area, or similar ones such as the Po Valley in Italy, we feel this information is essential for consideration.

The conclusions section is just a summary of main findings. It would be useful to add some discussion of needs for future work, in particular how some of the findings (such as dramatically incorrect emission inventories) could be further verified and eventually incorporated into better emission parameterizations. Is the ABL budget method a practical technique for grounding-truthing regional emissions on a model-relevant scale?

We have add two paragraphs to the conclusions in order to suggest further research that may build on the accomplishments of this study.

Specific Comments

P2/L27: Wolfe et al. (2015) is another relevant and recent citation.

Thank you, yes, we have added that reference at this point. We had already included it in our paper elsewhere but had neglected it here.

Equations 4-7 and discussion thereof: Seems inconsistent. For example, the surface/entrainment terms are given different symbols for O3 and water. And the entrainment flux sign seems wrong – a higher concentration of stuff in the ABL should give rise to a positive entrainment flux (stuff leaving the ABL) and a negative contribution to dX/dt. It might be more straightforward to show a generic budget equation for any scalar, and then discuss specific treatments for water, ozone and methane.

You are correct, equation 4 had a sign inconsistency from our other equations, and there was substantial inconsistency in the symbols we had used. We have more systematically applied consistent symbols for the scalar budget equations and corrected the sign mistake. In response to a perceived misconception apparent in the reviewer's comment, we further added some discussion to clarify the role of entrainment in the ABL budget equations. A higher ABL concentration with everything else fixed would give rise to a dilution of the boundary layer concentrations, and yes this drives a negative dC/dt. However, this is not due to "stuff leaving the ABL" as the reviewer states. Entrainment in an actively turbulent ABL is an irreversible mixing process that incorporates free tropospheric (FT) air into the ABL, not vice versa.  The positive scalar flux at the ABL top is the equivalent to a downward flux of concentration deficit (when the FT possesses a lower concentration), and we have explicitly stated that in the text now. We thank the reviewer for bringing this to our attention.

Page 8, Lines 16-22: suggest deleting.

Advice taken; we have removed these lines from the manuscript. We originally wanted to emphasize that in principal different scalars could be used in their respective budget equation to expose entrainment rates, i.e. water, ozone, or methane, and have made that point up front during the discussion of equations 4-7 as per reviewer's suggestions.

Eqn. 5: How are the BL concentrations determined for this calculation? Is it an average over the whole ABL, or just the upper portion? Same question for FT? Are uncertainties from this averaging (e.g. std of mean) propagated through to entrainment flux?

The scalar jump is determined from looking at vertical profiles and making the best eye judgment of the difference in concentrations between the top half of the ABL and the lowest ~100m of the FT. Often it is quite clear as can be seen in our example from fig. 7.  We have included a brief description of how these values are determined and their estimated uncertainties, which are like all the terms propagated through to the final results. The error analysis section (4) has been greatly expanded so this should be much clearer now.

P12/L7: how is this map generated? Is it an interpolation of ground site data? Please expound. Also, another way of stating the opposing O3 and NO2 advective terms is that Ox=O3+NO2 is conserved.

The NOx and O3 advective maps are interpolated to a 2D grid from aircraft data taken in the ABL. All data is corrected for the calculated mean regional time rate of change back to a common time stamp of 13:30. This has been more clearly explicated in the text.

As for the odd oxygen interpretation, we do not agree.  The gradient of ozone is an order of magnitude greater than that of $NO_2$. This is not simply a titration situation, but is intimately linked to rapid ozone production. We feel that the discussion of odd oxygen in this study would not serve to illuminate because it introduces a further unknown variable of the NOx emission rates. Also, we only had the $NO_2$ measurement on one single flight.

Section 3.2.3: These findings seem to suggest that NARR has serious flaws and should be adjusted, at least coarsely, to more accurately represent agricultural practices in some broad sense. A naïve question: would such issues impact the subsidence velocity derived from NARR?

We do not believe that large scale vertical motion would be all that susceptible to partitioning of surface heat fluxes among latent to sensible, but it certainly affects the convective activity and entrainment and boundary layer depths in the model. Subsidence is generally believed to be controlled by synoptic flow conditions. Although we do suspect that subsidence can modified a good bit due to mesoscale orography. A better representation of agricultural practices would lead to a better

estimate of the latent heat flux, which affects the partitioning in the surface buoyancy flux, and for a constant net radiation forcing this would lead to lower ABL heights for greater latent heat fluxes. This is why the NARR ABL depths are so much higher than measured, for instance.

Table 3: The third column is technically not a flux, but a flux divergence. Also, please give CH4 production in ppmv/h for easy comparison with other terms.

The third column is the entrainment flux contribution to the flux divergence. We report it that way to have it in comparable units to the other terms. But reporting the surface emission similarly would not make sense to us, as the units most people are familiar with are something like the chosen ones of Gigagrams per year. The CH4 production (surface emission) term is simply the numerical sum of the other columns, so we thought it would be redundant to see it in the same units.

Figure 9: is there any physical rationale behind a power-law fit?

The short answer is no. We know that the ozone chemistry is non-linear, and the simplest non-linear relationship is a power law.

Technical Comments

Fig. 2: Please label flight regions 1 and 2 as referenced in section 2.1.

We have changed the legend of Figure 2 to indicate the region numbers 1 and 2.

RASS is defined twice.

Got it.

P6/L32: delete ", which"

Deleted

P6/L35: "as per the Fundamental Theorem of Calculus" is a gratuitously pretentious statement.

We did not consider that such a foundational mathematical principle could be considered pretentious, but have eliminated the wording to protect the common reader.

Equations 1-3: subsidence is referred to as both $W(z_i)$ and $W$. Pick one.

Okay, thanks we will stick with just, W, with the implicit understanding that it can be a strong function of height.

P9, L13: delete "the 5 hour period of late morning to early afternoon from" P10/L17: delete "a remove of"

Done.

---

## Author Comment (AC6) · 27 Sep 2016

Note* All responses are in blue type.

Review: Observing Entrainment Mixing, Photochemical Ozone Production, and Regional Methane Emissions by Aircraft Using a Simple Mixed-Layer Model

This paper describes the design and execution of two flight experiments in the San Joaquin Valley of California to quantify entrainment rates and then uses these entrainment velocities to solve for: (a) ozone production rates, (b) methane emissions, and (c) evapotranspiration. The authors are attempting numerous things here, which makes the paper difficult to read and, at times, the results difficult understand. The work is interesting, but paper would benefit from better organization around a clear goal prior to publication. Adding clarity may be as simple as removing the excessive inessential detail.

General comments:

The Introduction should be reorganized to better frame the work. Some specific issues are as follows.
In paragraph 2, the text does not define "tracer method" or "budget of the inversion base height" when describing what is done in the forthcoming analysis. This makes it difficult for the reader to know what is done here and how this work is different from past work.

We have added some clarification clauses to describe these methodologies, but exact details have to be postponed to the method descriptions of Section 2.

The sentence, "by way of targeted airborne campaigns we are able to probe the regional ABL vertically and horizontally and calculate entrainment rates and mesoscale advection," seems key, but is placed awkwardly in the middle of paragraph 3.

This statement is made after introducing the concepts of entrainment and advection, and therefore does not seem awkward in its placement to us. We have attempted to make a more clarion statement of the paper's overarching goal at the end of paragraph 3, keeping in mind that positional emphasis is typically carried by the end sentence of a paragraph (The Elements of Style, by Strunk & White [1999]):

> *The central goal of the work presented here is to show how, by way of targeted small-scale airborne campaigns, it is possible to probe the regional ABL vertically and horizontally to calculate entrainment rates and mesoscale advection, and thereby shed light on all of the processes that change the concentrations of trace gases in the boundary layer throughout the day. This methodology thereby reveals the quantitative origins of chemical constituents measured in near-surface air, by comparing direct observations of all but one of the leading terms of the scalar budget equation, and inferring the unknown term as a residual.*

The fourth paragraph returns to the idea of scalar budgeting, but still does not define, instead suggesting I should already be familiar with the concept (done through the particular way the references are discussed).

*We have defined a scalar budget in an added subordinate clause in the second paragraph, and a new sentence at the end of the third paragraph as per earlier suggestions. Then we devote the entirety of Section 2.7 to defining exactly what the methodology is. We do not see how to further clarify the technique in the introduction without burdening the section with excessive detail.*

While I agree with the content in paragraph 5, this paper is not actually about, "better understand[ing] the diurnal behavior of the wintertime boundary layer in the San Joaquin Valley."

*We think that reporting observed entrainment rates in the winter, which have never been reported, does in fact help to better understand the ABL's diurnal behavior.*

The discussion in paragraph 6 should more relevant to the analysis performed. For example, the paper never significantly discusses PM, but investigates ozone production, methane emissions, and evapotranspiration. While there is some text on ozone and drought here, methane is absent entirely.

*We have added a concluding sentence to this paragraph that helps to establish the importance of the work:*

> *Entrainment aloft becomes an even more important factor during stagnant conditions in the SJV because it represents the principal mode of ventilating the air pollutants in the ABL, and therefore its quantification is crucial to predicting the intensity and duration of an air quality episode.*

*Although the work does not explicitly address PM issues, the results are directly applicable to the wintertime PM problem in the SJV and we hope will be used by others working on the DISCOVER-AQ data set. Also, because methane is not directly an air quality concern, we leave it out of this paragraph. We have removed a couple of sentences in the hopes that they might be considered "excessive inessential details."*

The last paragraph presents an outline of the paper, but the preceding text has not setup these goals, nor does the outline mention the ozone production, methane emission, or evapotranspiration applications.

*We have expanded the outline paragraph in an attempt to state the goals of our work more clearly, as per the reviewer's earlier suggestion.*

Most of Section 2.1 is irrelevant. The authors should relate the descriptive information directly back to their analysis and delete superfluous detail.

We have condensed much of the information originally presented in Section 2.1 as it was also suggested by reviewer 2. However, we disagree that this discussion of the dynamic environment is irrelevant. We chose to include a clear survey of mountain-valley dynamics to set the stage for this unique mesoscale environment in which the experiments took place and because we do not find such a concise description anywhere in the extant literature. This dynamic complexity lies at the heart of why the region endures some of the poorest air quality in the nation. For others working on recalcitrant air quality issues in this area, or similar ones such as the Po Valley in Italy, we feel this information is essential for consideration.

Sections 2.6 and 2.7 should be framed around what was done here, rather than as done currently, as a general discussion of the two methods using the author's dataset as an example. The last sentence of Section 2.7, "ultimately the approach using the budget of boundary layer inversion height, outlined in Section 2.6 was taken to calculate the entrainment rate," should be given to the reader up front. Additionally, the last paragraph in 2.7 is described almost narratively of how the analysis was done. Please reorder such that results are presented to convey the logic of the analysis to the reader.

We have restructured/rewritten Section 2.7 to better coordinate the general discussion of the scalar budget equations with how they were used in these experiments.

What are the results for Ox, as opposed to O3 and NO2 separately? Use of P(Ox) would be especially important in the wintertime and better suited for a winter/summer comparison. Secondly, has wintertime P(O3) been found to be NOx-limited also? That seems unlikely; please clarify.

Unfortunately, we did not have measurements of $NO_2$ save for one single flight, and therefore were not able to perform a budget of odd oxygen.

- Yes, the results presented in Fig. 9 indicate that P(O3) is NOx-limited in the wintertime, but the inference is not strong given the limited spread in VOC:NOx ratio, and the uncertainties in using CH4 as a general VOC proxy. Nevertheless, we feel the result is worth presenting, especially since very little is known about winter O3 production because it is not often considered.

Broadly, the outline of the paper is to compute the entrainment rate and then use this rate to explore three things: (a) ozone production rates, (b) methane emissions, and (c) water. Adding text or a dedicated section after discussion of the three studies, but prior to the Conclusion, that ties everything back together would do two valuable things. First, it would clarify the narrative and logic of the paper, and second, it would reinforce the significance of the work.

We have attempted to tie everything together more clearly throughout the revised manuscript and thus do not see the value in repeating this before doing so again in the conclusions.

Specific comments:

Page 2, lines 3–4: Citation needed on, "this mixing tends to be a significant contributor to the ABL budget of the scalar."

Stull [1990], Arellano et al. [2011], Lehning et al. [1998].

Page 3, lines 17–18: Should this be 105 exceedances "per year"?

We have eliminated this statement as non-essential.

Page 7, line 7: w(e) is not defined in the text (it is instead defined on page 8, line 23).

Defined in both places now.

Page 10, lines 18–20: What is the evidence for: "For the purposes of estimating regional source strengths or regional in situ photochemistry, we suggest that the more pertinent mixing process is the dilution of the anthropogenically influenced ABL air mass by the more global 'baseline' FT air."

This is more of a conjecture, claiming that it is the ABL growth rate after its initial 'encroachment' through the morning's residual layer that is key in understanding regional chemistry and surface emissions because the residual layer tends to be made up of mostly recycled air from the region. Of course, in principle, the budgets should still hold during the more rapid growth of the morning ABL, but they become more difficult to accurately measure due to the greater presence of transients and inhomogeneities. We do not feel this detail should be introduced into the manuscript because it is somewhat tangential as we did not perform the budget analysis in the morning hours, and it would not make sense to anyway because of the low $O_3$ production at high solar zenith angles, which does not impact the afternoon $O_3$ maximum very significantly.

Page 11, lines 34–35: How is this shown in Fig. 7: "the importance of entrainment mixing on an ozone exceedance day."

It is shown in the subsequent discussion where the jumps observed in Fig. 7 are used to estimate a time rate of change of $O_3$ and $NO_2$ concentrations due to entrainment dilution.

Page 12, lines 35–36: It is difficult to see that methane is an appropriate proxy for total VOC. Even if dairies and gas production are the dominant source of VOCs, what matters more is that the drivers of methane emission match the drivers of the other VOC, which might not be true even if the sources are the same.

As discussed in Section 3.2.2 the majority of methane in both studies are believed to be

associated with fossil fuel extraction and dairy operations. The studies of Gentner et al. [2014] and Pusede et al. [2014] indicate that methane is fairly well correlated with alcohols (which have strong dairy sources), higher alkanes (natural gas), and CO (other anthropogenic activities.) While we acknowledge that methane is a somewhat crude tracer of reactive VOC, we present the results because there is a suggestive relationship with our inferred ozone production rates that is consistent with past studies of the ozone production regime.

Page 13, lines 3–5: Can an estimate of the uncertainty be given?

We have included an average uncertainty estimate from our experimental results to better frame the comparison, and have done so in all of the Tables as well.  There is no estimate of uncertainty in P(O3) made by Pusede et al. (2014).

Section 4: I recommend moving Section 4 to precede Sections 3.2.1–3.2.3.

We feel that a discussion of the errors in the measurements specifics is best delayed until the details of the experimental results are related, so we have kept Section 4 after Section 3, but we have expanded it considerably to make clear exactly how our errors have been treated in our results.

---

## Author Response (AR2)

Title: if there is concern about the word "model" causing confusion, it could be changed to "framework."

-Suggestion taken. Thank you very much for the suggestion.

Abstract: please include uncertainties for CH4 emissions here.

-Done.

Page 4/Line 10: "Sierra Nevada mountains."

-Fixed.

Section 3.2.2: removing/modifying the data for Fig. 10 seems sketchy. It might be better to show that data in the plot and not use it for the fit (which I think you justified in the text).

-The reason for leaving out the data is justified in the text and in fact one of the data points is from a slightly negative emission flight results, which is impossible to show on a logarithmic axis. We feel that adding the points back in, despite not being a part of the fit, would be more confusing than is worth it.

Page 15/Line 16: standard error should be normalized by square root of number of points.

-Fixed

Figure 4 caption: please distinguish solid and dashed lines.

-Done.

Figure 9: fit seems unnecessary. If there is a trend, the data will stand on its own.

-The power law fit here is to represent that fact that ozone chemistry is non-linear and we chose to fit with a power law because it is the simplest non-linear relationship. We feel that the approximate power of the fit might have meaning to some researchers more closely familiar with non-linear ozone chemistry.

Table 2: The relative contributions of the chemical and dynamical terms seems to be the driving theme of the paper, and this does come out in an average sense in the text, but it is hard to glean the full range of variability

and relative magnitudes from a dense block of numbers. You might consider plotting the rates as a set of bar charts, or in some other visual fashion. I realize that this is a major change and may break the paper's symmetry of 3 similar tables, but it could be the figure that drives the whole point home. This could be done for all three tables in fact, and the tables themselves moved to a supplement for those who wanted the actual numbers and uncertainties. Just a thought.

-We took the suggestion and found that a new figure which shows the budget terms averaged for each respective flight mission to be quite helpful in ascertaining the relationships discussed in the text. We therefore added the three tables to a supplement. We thank you very much for the excellent suggestion.

[revised manuscript text omitted]